# FcαRI co-stimulation converts human intestinal CD103[+] dendritic cells into pro-inflammatory cells through glycolytic reprogramming

Ivo S. Hansen[1,2], Lisette Krabbendam[2], Jochem H. Bernink[2], Fabricio Loayza-Puch [3], Willianne Hoepel[1,2], Johan A. van Burgsteden[1,2], Elsa C. Kuijper[4], Christianne J. Buskens[5], Willem A. Bemelman[5], Sebastiaan A.J. Zaat[6], Reuven Agami[3], Gestur Vidarsson [7], Gijs R. van den Brink[8], Esther C. de Jong[2], Manon E. Wildenberg[8], Dominique L.P. Baeten[1,2], Bart Everts[4] & Jeroen den Dunnen[1,2]

CD103[+] dendritic cells (DC) are crucial for regulation of intestinal tolerance in humans. However, upon infection of the lamina propria this tolerogenic response is converted to an inflammatory response. Here we show that immunoglobulin A (IgA) immune complexes (IgA-IC), which are present after bacterial infection of the lamina propria, are important for the induction of inflammation by the human CD103[+]SIRPα[+] DC subset. IgA-IC, by recognition through FcαRI, selectively amplify the production of proinflammatory cytokines TNF, IL-1β and IL-23 by human CD103[+] DCs. These cells then enhance inflammation by promoting Th17 responses and activating human intestinal innate lymphoid cells 3. Moreover, FcαRI-induced cytokine production is orchestrated via upregulation of cytokine translation and caspase-1 activation, which is dependent on glycolytic reprogramming mediated by kinases Syk, PI3K and TBK1-IKKε. Our data suggest that the formation of IgA-IC in the human intestine provides an environmental cue for the conversion of a tolerogenic to an inflammatory response.

[1] Amsterdam Rheumatology and Immunology Centre, Department of Clinical Immunology and Rheumatology, Academic Medical Centre, University of Amsterdam, Meibergdreef 9, 1105 AZ Amsterdam, The Netherlands. [2] Department of Experimental Immunology, Academic Medical Centre, University of Amsterdam, Meibergdreef 9, 1105 AZ Amsterdam, The Netherlands. [3] Division of Oncogenomics, Netherlands Cancer Institute, Plesmanlaan 121, 1066 CX Amsterdam, The Netherlands. [4] Department of Parasitology, Leiden University Medical Centre, University of Leiden, Albinusdreef 2, 2333 ZA Leiden, The Netherlands. [5] Department of Surgery, Academic Medical Centre, University of Amsterdam, Meibergdreef 9, 1105 AZ Amsterdam, The Netherlands. [6] Department of Medical Microbiology, Academic Medical Centre, University of Amsterdam, Meibergdreef 9, 1105 AZ Amsterdam, The Netherlands. [7] Department of Experimental Immunohematology, Sanquin Research, University of Amsterdam, Plesmanlaan 125, 1066 CX Amsterdam, The Netherlands. [8] Tytgat Institute for Liver and Intestinal Research and Department of Gastroenterology and Hepatology, Academic Medical Centre, University of Amsterdam, Meibergdreef 9, 1105 AZ Amsterdam, The Netherlands. Correspondence and requests for materials should be addressed to J.d.D. (email: j.dendunnen@amc.nl)

The human intestine is constantly exposed to a high load of foreign antigens, between which the intestinal immune system must discriminate and provide tolerance to commensal micro-organisms and dietary antigens, while inducing a protective immune response to invading pathogens. Crucial for maintaining homeostasis are a subset of intestinal dendritic cells (DC) that are characterized by expression of the integrin CD103 (αE integrin), which are scattered throughout the lamina propria of the intestine[1–3]. CD103[+] DCs propagate tolerance through their capacity to migrate from the lamina propria to draining mesenteric lymph nodes[4,5] to induce gut-homing regulatory T cells[6,7]. In addition, CD103[+] DCs orchestrate intestinal homeostasis via activation of T helper cells (Th) and innate lymphoid cells (ILC)[1,8,9]. Reports indicate that human intestinal CD103[+] DCs can be further divided into different subsets with distinct immunological functions, which is mainly based on the expression of SIRPα[10,11]. For example, while the SIRPα[+] subset is more specialized in activating Th17 and type 3 ILC (ILC3), the SIRPα[-] subset is more efficient at inducing Th1 responses[11].

In contrast to homeostatic conditions, when pathogens enter the lamina propria upon intestinal infection, the tolerogenic response induced by CD103[+] DCs should be converted to a proinflammatory response to protect the host. In most tissues, DCs induce proinflammatory responses directly in response to microbial structures, which are detected by distinct families of pattern-recognition receptors (PRR) such as Toll-like receptors (TLR)[12], NOD-like receptors (NLR)[13], and C-type lectins (CLR)[14]. However, PRR stimulation of intestinal CD103[+] DCs already occurs under homeostatic conditions because of the high load of PRR-activating microbial structures in the intestinal lumen that reach the CD103[+] DCs through several transfer mechanisms, which include direct luminal sampling[15], antigen sampling via Goblet cells[16], and gap junction transfer from CX3CR1[+] macrophages[17]. Although PRR stimulation enhances CD103[+] DC migration to mesenteric lymph nodes[18], PRR stimulation of intestinal CD103[+] DCs is strongly associated with induction of tolerance through aforementioned mechanisms[1,3,6,19]. Different hypotheses have emerged to explain how the intestinal immune system converts the tolerogenic response induced by CD103[+] DCs to an inflammatory response upon infection. These include activation of other phagocytic cell populations, such as CD103[-] DCs or CX3CR1[+] macrophages[2,10], thereby overruling the tolerogenic response of CD103[+] DCs[3]. An alternative possibility is that intestinal CD103[+] DCs themselves switch from a tolerogenic to an inflammatory phenotype by sensing the altered environment of the infected mucosa[3,20], e.g., by detection of a 'second signal' in addition to the continuous exposure to microbial antigens. Yet, currently it is still unclear whether or how CD103[+] DCs can convert from immune tolerant to proinflammatory cells.

Immunoglobulin A (IgA) is the major antibody isotype in humans and is expressed particularly high in the intestine[21]. IgA is produced in the lamina propria by local plasma cells and is subsequently transported through epithelial cells into the intestinal lumen[22]. It has become clear that the intestinal IgA repertoire is tailored to recognize an individual's microbiota[21,23], and that IgA particularly opsonizes bacterial species that are colitogenic[24]. While under homeostatic conditions the intestinal lumen, but not the lamina propria, contains IgA opsonized bacteria, under pathological conditions such as infection invading IgA opsonized micro-organisms enter the lamina propria[22], thereby introducing high quantities of IgA immune complexes (IgA-IC) to be recognized by the intestinal immune system.

Here, we provide evidence that the presence of IgA-IC delivers a second signal to human CD103[+]SIRPα[+] intestinal DCs that strongly promotes proinflammatory responses by these otherwise tolerogenic cells. We demonstrate that IgA opsonization of bacteria amplifies the production of proinflammatory cytokines such as TNF, IL-1β, and IL-23 by human CD103[+] DCs. This amplification of proinflammatory cytokine production is induced by cross-talk of IgA receptor Fc alpha receptor I (FcαRI, or CD89) with TLRs, NLRs and CLRs. Furthermore, we demonstrate that this conversion of CD103[+] DCs from tolerogenic to inflammatory cells is dependent on FcαRI-induced glycolytic reprogramming, which is mediated by signalling through kinases Syk, PI3K, and TBK1-IKKε causing an increase in gene translation and caspase-1 activation.

## Results

**IgA opsonization enhances inflammatory cytokine production.** Bacteria that penetrate the epithelial layer of the gut introduce IgA-IC into the lamina propria as a result of their opsonization with IgA. Here, we set out to investigate whether IgA opsonization of bacteria affects the response of human intestinal CD103[+] DCs. *Staphylococcus aureus* was used as a model bacterium, of which opsonization was verified as previously described[25]. To investigate the response of CD103[+] DCs, we made use of a human monocyte-derived in vitro model that closely resembles primary human CD103[+] DCs in both phenotype and function[9,26], in which CD103[+] DCs are generated by priming of monocytes with retinoic acid. The phenotype of these cells is depicted in Supplementary Fig. 1, which shows expression of various DC markers, absence of monocyte/macrophage marker CD14, and homogeneous expression of CD103. The in vitro CD103[+] DCs closely resembles the SIRPα-expressing subset of CD103[+] primary human intestinal DCs, characterized by expression of SIRPα (CD172a), BDCA1 (CD1c), intermediate expression of BDCA3 (CD141), and absence of CLEC9A (CD370) (Supplementary Fig. 1)[10,11]. Similar to primary CD103[+]SIRPα[+] DCs, the in vitro CD103[+] DCs have a high expression of CD86 but not of other maturation markers such as CD80 and CD83, indicating that the high expression of CD86 may be specific for this DC subset[10]. Combined, these data indicate that the in vitro CD103[+] DC model resembles CD103[+]SIRPα[+] intestinal DCs.

To assess the effect of IgA opsonization of bacteria we co-cultured CD103[+] DCs with live IgA opsonized and non-opsonized *S. aureus* for 24 h and subsequently harvested the supernatant for quantification of cytokine levels. Strikingly, IgA opsonization of bacteria enhanced the production of proinflammatory cytokines TNF, IL-6, and IL-23 (Fig. 1). In contrast, IgA opsonization of bacteria had no effect on cytokine production by conventional human monocyte-derived DCs (moDCs) (Fig. 1), indicating that the amplification of cytokine production is specific for CD103[+] DCs. These results suggest that IgA opsonization of bacteria amplify the production of proinflammatory cytokines by CD103[+] DCs.

**IgA-IC synergize with various PRRs.** While TLR stimulation of CD103[+] DCs has been described to continuously occur under homeostatic conditions, the presence of IgA-IC mainly occurs under certain conditions such as bacterial invasion, and therefore may function as an environmental cue to indicate danger. To investigate whether IgA-IC induce cytokine production directly, we stimulated CD103[+] DCs with IgA-IC, generated by coating purified human IgA on high-affinity culture plates. First, we verified that potential endotoxin levels were too low to affect cytokine production (Supplementary Fig. 2). Remarkably, stimulation with IgA-ICs induced very little amounts of cytokines (Fig. 2a). Stimulation with TLR2/1-specific ligand Pam3CSK4 induced low but detectable levels of cytokines TNF, IL-6, and IL-23, as well as anti-inflammatory cytokine IL-10 (Fig. 2a). Strikingly, co-stimulation with Pam3CSK4 and IgA-IC strongly and

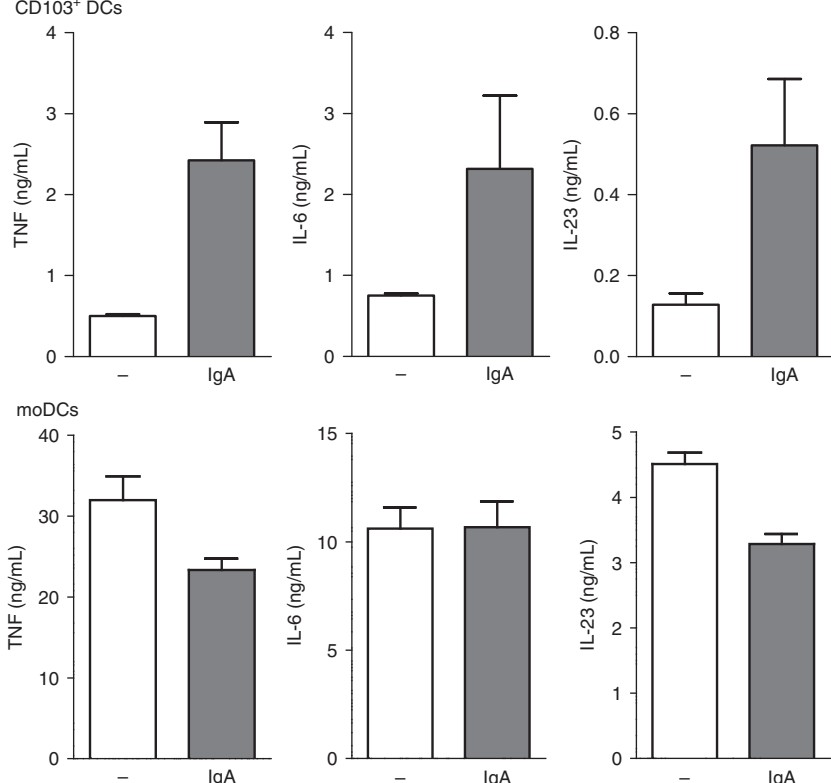

**Fig. 1** IgA opsonization of bacteria enhances production of proinflammatory cytokines by CD103[+] DCs. CD103[+] dendritic cells (DC) or moDCs were stimulated with 10 MOI *S. aureus*, which were untreated or opsonized with 5 mg/mL serum IgA for 1 h and washed. Experiments were performed in triplicate. After 24 h cytokine levels were analysed using ELISA, mean + s.e.m. Representative example of three experiments with different donors

synergistically amplified the production of proinflammatory cytokines TNF, IL-1β, and IL-23, while the production of IL-6 and IL-10 was moderately increased (representative example Fig. 2a; multiple donors Fig. 2b). In contrast to CD103[+] DCs, co-stimulation of Pam3CSK4 with IgA-IC had very little effect on proinflammatory cytokine production by conventional moDCs (representative example Fig. 2c; multiple donors Supplementary Fig. 3), suggesting that this synergistic amplification is restricted to CD103[+] DCs. We verified these findings with primary human CD103[+] DCs isolated from surgically removed intestine (Supplementary Fig. 4). Similar to the in vitro model, co-stimulation of Pam3CSK4 with IgA-IC increased TNF production by ex vivo CD103[+] DCs (Fig. 2d), as well as by CD103[−] DCs (Supplementary Fig. 5). These data indicate that stimulation with IgA-IC strongly amplifies the production of proinflammatory cytokines by CD103[+] DCs through synergy with TLRs.

Notably, stimulation with IgA-IC also amplified proinflammatory cytokine production induced by TLR2 alone (LTA) and TLR4 (LPS) (Fig. 2e), two main members of bacteria-sensing TLRs. In addition, IgA-IC co-stimulation upregulated proinflammatory cytokine production induced by TLR7/8 (CL097), but had little effect on TLR3 (PolyIC), both PRRs important in virus recognition. Stimulation with IgA-IC did upregulate TNF production by NOD2 (PGN and MDP), belonging to the family of NLRs, as well as Dectin-1 (curdlan), a member of the CLR family that is important in the recognition of fungi[14]. Alternatively, we tested whether TLR-induced cytokine production by CD103[+] DCs is also amplified by immunoglobulin G (IgG)-IC, which indeed induced a similar response as IgA-IC (Supplementary Fig. 6). Thus, IgA-IC and IgG-IC stimulation of CD103[+]

DCs amplifies proinflammatory cytokine production induced by a diverse range of PRR families.

Taken together, these data indicate that while exposure to microbial components induces little inflammation, co-stimulation with IgA-IC acts as a second signal that strongly promotes a proinflammatory response by intestinal CD103[+] DCs, which is characterized by highly elevated production of TNF, IL-1β, and IL-23.

**IgA1 and IgA2 amplify cytokine production through FcαRI.** Next, we set out to identify the receptor responsible for the IgA-induced upregulation of proinflammatory cytokines by CD103[+] DCs. One of the main IgA receptors in humans is FcαRI[27]. FcαRI expression was low but detectable on in vitro generated CD103[+] DCs (Fig. 3a), which was comparable to expression by in vitro generated macrophages (Supplementary Fig. 7)[28]. In addition, primary CD103[+] DCs isolated from human intestine have markedly higher *FCAR* expression compared to primary BDCA1[+] DCs isolated from blood (comparison with different cell types is depicted in Supplementary Fig. 8). To determine whether FcαRI is responsible for the synergistic upregulation of cytokines by CD103[+] DCs upon co-stimulation with IgA-IC, we blocked FcαRI using a specific antagonistic antibody and analysed cytokine production. Blocking of FcαRI almost completely inhibited IgA-IC-induced upregulation of TNF, IL-1β, IL-6, and IL-23 (Fig. 3b). These data identify FcαRI as the main receptor for IgA-IC-induced amplification of proinflammatory cytokines by CD103[+] DCs.

IgA comprises two subclasses, IgA1 and IgA2, of which the ratio differs depending on the intestinal location. IgA1

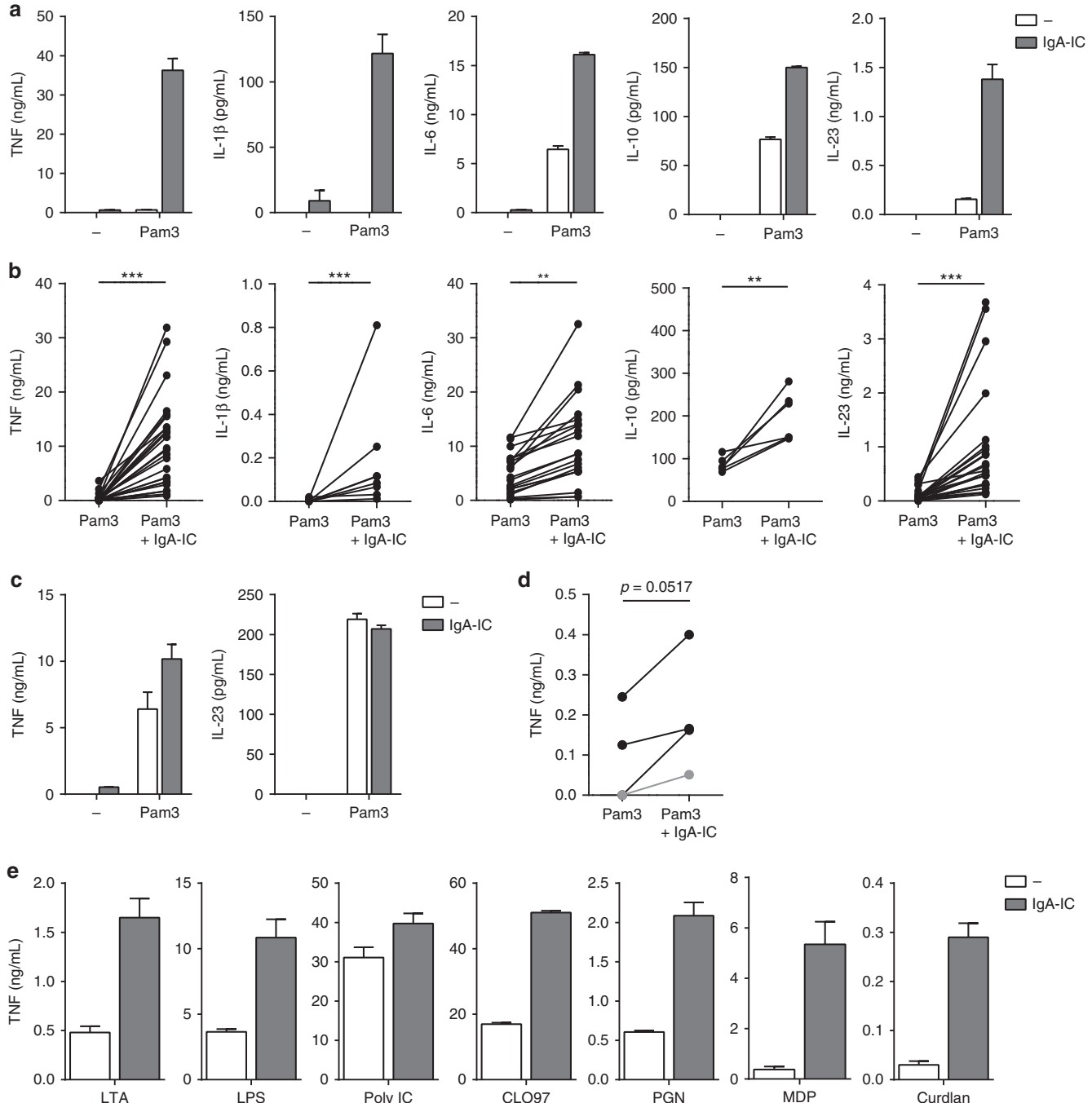

**Fig. 2** IgA-IC promote proinflammatory cytokine production through synergy with various PRRs. **a** CD103[+] dendritic cells (DC) were stimulated with Pam3CSK4 (Pam3), IgA immune complexes (IgA-IC), or a combination. **b** Cytokine production by CD103[+] DCs stimulated with Pam3CSK4 or Pam3CSK4 combined with IgA-IC. Each pair of dots represents one donor. *$p < 0.05$, **$p < 0.01$, ***$p < 0.001$, Mann–Whitney $U$-test. **c** Cytokine production by moDCs stimulated with Pam3CSK4, IgA-IC, or a combination. **d** CD103[+] DCs isolated from surgically resected intestine (1× ileum in grey, 3× colon in black) were stimulated with Pam3CSK4 or Pam3CSK4 combined with IgA-IC. Each pair of dots represents one donor. *$p < 0.05$. Student's $t$-test. **e** Cytokine production by CD103[+] DCs stimulated with different pathogen-associated molecular patterns (PAMP) alone or combined with IgA-IC. Stimulated receptors were TLR2 (LTA), TLR4 (LPS), TLR3 (Poly IC), TLR7/8 (CLO97), TLR2/NOD2 (PGN), NOD2 (MDP), and Dectin-1 (Curdlan). Experiments were performed in triplicate. After 24 h cytokine levels were analysed using ELISA, mean + s.e.m. Representative example of five (**a**) or three (**c** and **e**) experiments using different donors

predominates in proximal gut mucosa, while IgA2 is more abundant in distal gut mucosa[29]. To investigate whether there are differences between the two isotypes in their capacity to amplify cytokine responses, we compared the effect of co-stimulation of CD103[+] DCs with IgA1-IC and IgA2-IC. The two isotypes amplified cytokine production in a similar manner, characterized by strongly increased production of TNF, IL-1β, and IL-23 (Fig. 3c), indicating an equal proinflammatory response to both isotypes.

Taken together, these data indicate that IgA1 and IgA2 immune complexes equally promote cytokine production by CD103[+] DCs through FcαRI.

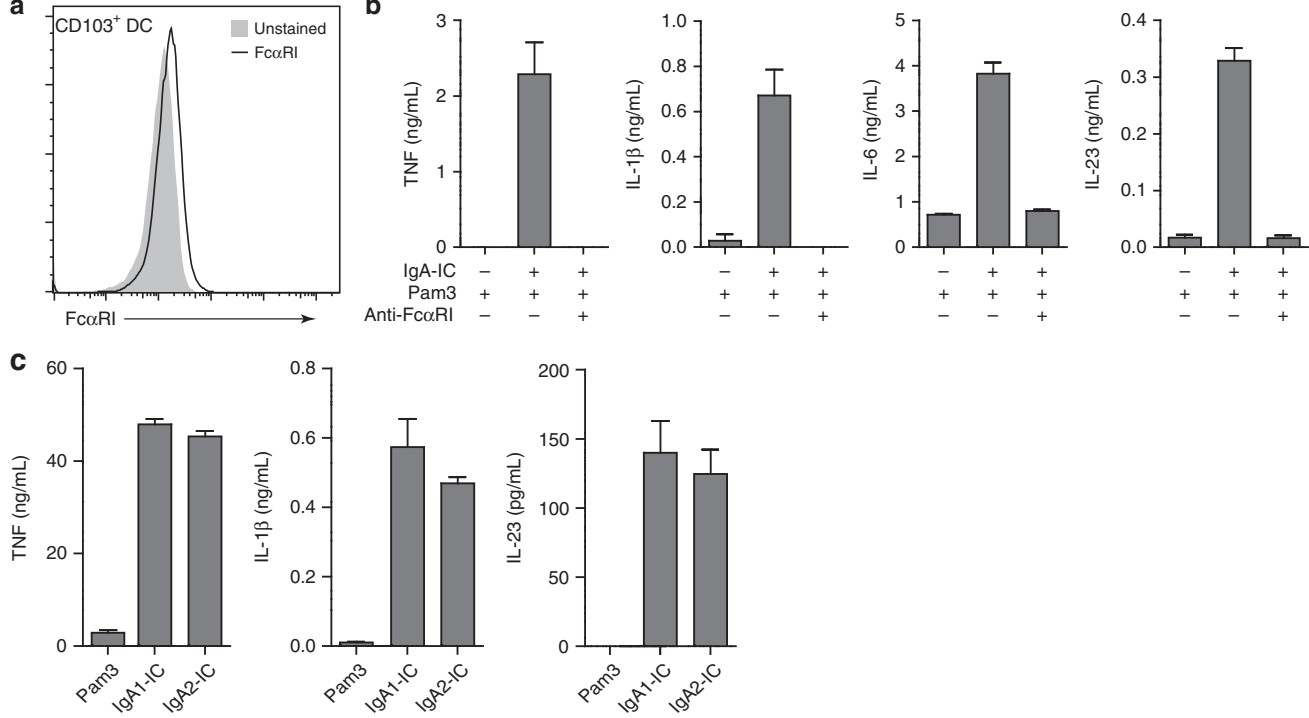

**Fig. 3** Complexed IgA1 and IgA2 amplify proinflammatory cytokine production through FcαRI. **a** FcαRI expression on unstimulated CD103[+] dendritic cells (DC) was analysed using flow cytometry. Light grey histogram indicates background staining. **b** Prior to stimulation, CD103[+] DCs were incubated with a blocking antibody against FcαRI and stimulated with Pam3CSK4 (Pam3) alone or Pam3CSK4 together with IgA immune complexes (IgA-IC). **c** Cytokine production by CD103[+] DCs stimulated with Pam3CSK4 alone or in combination with IgA1-IC or IgA2-IC. Experiments **b** and **c** were performed in triplicate. After 24 h cytokine levels were analysed using ELISA, mean + s.e.m. Representative example (**a**–**c**) of three experiments using different donors

**Co-stimulation promotes Th17 responses and activates ILC3.** An important mean by which DCs control intestinal inflammation is by migrating to mesenteric lymph nodes to activate and promote the differentiation of distinct effector T helper cell subsets[3]. One of the key processes in orchestrating protective intestinal immunity is the induction of Th17 responses[30,31]. Human Th17 polarization is thought to be dependent on cytokines IL-1β, IL-6, IL-23, and TNF[32–36], which were all upregulated upon co-stimulation of CD103[+] DCs with IgA-IC. To investigate whether IgA-ICs affect T helper cell activation induced by CD103[+] DCs, we co-cultured stimulated CD103[+] DCs with allogeneic CD4[+] T cells and quantitatively determined the secretion of hallmark cytokines IL-17 (Th17), IFN-γ (Th1), IL-13 (Th2), and IL-10. Co-stimulation of Pam3CSK4 with IgA-IC upregulated IL-17 production, but hardly affected the production of IFN-γ or IL-13, while production of anti-inflammatory IL-10 was suppressed (Fig. 4a). These data indicate that co-stimulation of CD103[+] DCs with IgA-IC selectively promote Th17 responses.

In addition to T cells, it has become clear that innate lymphoid cells (ILC) are also essential in the orchestration of inflammation and tissue repair in the intestine[37]. Of the different ILC subsets in the intestine, ILC3 represent the predominant subset which is involved in host defence and wound-healing through the production of IL-22[8]. Since activation of ILCs critically depends on cytokines, we assessed the effect of FcαRI-induced cytokine production by CD103[+] DCs on ILC3 activation through transfer of CD103[+] DC supernatant. Co-stimulation of CD103[+] DCs with IgA-IC strongly enhanced the expression of ILC3-derived IL-22 on mRNA (Fig. 4b) and on protein level (Fig. 4c), but did not induce detectable levels of *IL17A* or *IL17F* (Fig. 4b). Furthermore, activation of ILC3 was linked to enhanced

expression of the ILC3-associated transcription factor *RORC* (Fig. 4b).

Taken together, these data suggest that FcαRI co-stimulation of CD103[+] DCs skews intestinal immune responses by both promoting Th17 responses and by activating ILC3, which are stimulated to produce IL-22.

**Enhanced gene translation and caspase-1 activation by FcαRI.** Modulation of cytokine production can be orchestrated at different levels. To determine whether FcαRI-induced upregulation of cytokines is regulated at the transcriptional level, we analysed mRNA expression of the genes of interest. However, co-stimulation of FcαRI had very little effect on TLR-induced transcription of *TNF*, *IL1B*, *IL6*, and *IL23A*, nor on other genes such as *CXCL8*, *IL10*, *IL12A*, and *IL12B* (Fig. 5a). While we observed a trend towards increased mRNA expression 6 h after stimulation, possibly indicating an increase in mRNA stability, further evaluation found that stability of the existing transcripts by TLR stimulation was not affected by IgA-IC stimulation (Fig. 5b).

Next, we assessed whether the amplified cytokine production by FcαRI co-stimulation was regulated at the level of gene translation. We evaluated changes in cytokine gene translation by measuring the relative amount of ribosomes bound to the mRNA of the upregulated genes. Sucrose gradient centrifugation of lysates from stimulated cells was used to separate ribosome-free mRNA from mRNA bound to ribosome chains, known as polysomes. Strikingly, co-stimulation of Pam3CSK4 with IgA-IC increased *TNF* translation initiation as well as polysome formation (Fig. 5c), suggesting that FcαRI co-stimulation amplifies the production of proinflammatory cytokines by CD103[+] DCs by enhancing gene translation.

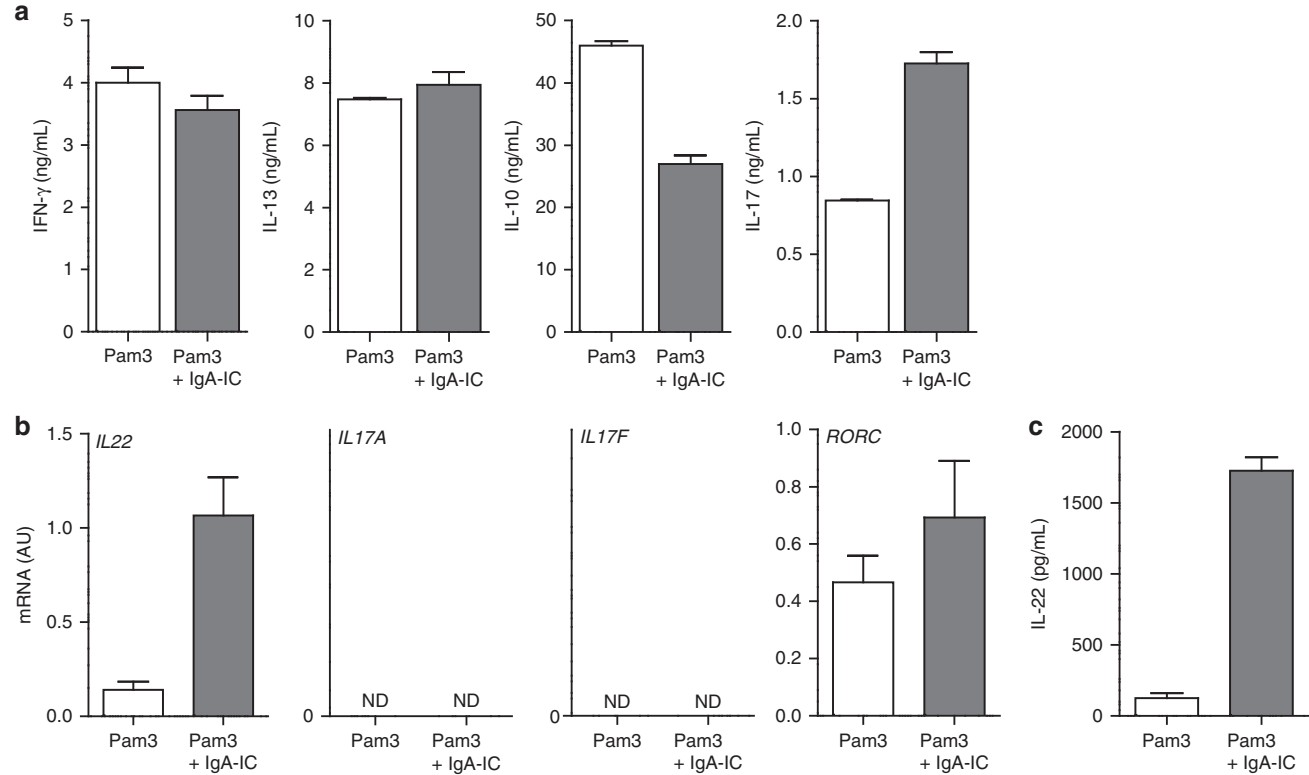

**Fig. 4** Co-stimulation of CD103[+] DCs with IgA-IC promotes Th17 responses and activates intestinal ILC3. **a** CD103[+] dendritic cells (DC) were stimulated with Pam3CSK4 (Pam3) alone or Pam3CSK4 together with IgA immune complexes (IgA-IC) and co-cultured with CD4[+] T Cells. After T-cell outgrowth, resting cells were re-stimulated and after 24 h supernatant was assayed with ELISA, mean + s.e.m. Experiments were performed in triplicate. **b**, **c** Supernatant from CD103[+] DCs which were stimulated with Pam3CSK4 or Pam3CSK4 with IgA-IC was transferred onto human intestinal ILC3 and cultured for 7 days and analysed for mRNA expression (**b**) or protein expression (**c**). ND, not detected. Experiments were performed in triplicate. mRNA expression of indicated genes were assayed using qPCR (normalized to *GAPDH* expression) mean + s.e.m. Supernatant was analysed using ELISA, mean + s.e.m. Representative example of three (**a**, **b**) or two (**c**) experiments using different donors

Remarkably, co-stimulation of CD103[+] DCs with IgA-IC did not enhance translation initiation or polysome formation of *IL1B* (Supplementary Fig. 9), suggesting a mechanism other than gene transcription or gene translation is responsible for increased IL-1β production. Since pro-IL-1β needs to be processed post-translationally to functional IL-1β[38], we examined caspase-1 activity after (co-) stimulation of CD103[+] DCs using caspase-1 binding compound FAM-YVAD-FMK. While Pam3CSK4 stimulation had little effect, IgA-IC stimulation activated caspase-1, which was independent of TLR co-stimulation (Fig. 5d). In addition, caspase-1 inhibition by specific inhibitor Ac-YVAD-cmk inhibited IL-1β production induced by FcαRI-TLR co-stimulation without affecting TNF production (Supplementary Fig. 10), indicating that caspase-1 activation is required for FcαRI-induced IL-1β production.

Combined, these data indicate that FcαRI-TLR co-stimulation amplifies the production of proinflammatory cytokines by CD103[+] DCs by both upregulating gene translation and by activating caspase-1.

**FcαRI drives glycolytic reprogramming by Syk, PI3K, and TBK1.** Subsequently, we set out to identify the underlying molecular mechanisms of FcαRI-induced amplification of cytokine gene translation. Since mechanistic target of rapamycin (mTOR) is considered a master regulator of translation in many cells, we assessed whether mTOR inhibition affected cytokine production by IgA-IC/TLR co-stimulation. However, inhibition of mTORC1 and mTORC2 using inhibitor Torin had no effect on FcαRI-induced cytokine production (Supplementary Fig. 11a). Inhibition of mTORC1 using rapamycin moderately suppressed

TLR-induced cytokine production (Supplementary Fig. 11b), but did not specifically affect the upregulation of cytokines upon co-stimulation with IgA-IC (Supplementary Fig. 11c), suggesting alternative mechanisms. It has become apparent that for induction of proinflammatory responses intracellular metabolic changes in immune cells are of crucial importance, which is collectively referred to as metabolic reprogramming[39]. TLR agonists were found to promote a rapid increase in glycolytic flux in murine DCs, which serves an essential function in supporting the de novo synthesis of fatty acids for the expansion of the endoplasmic reticulum required for translation of proteins associated with DC activation[40]. To determine whether a similar process was operating in CD103[+] DCs after FcαRI stimulation, we stimulated CD103[+] DCs with IgA-IC and analysed for acute real-time changes in rates of extracellular acidification (ECAR), as a measure of lactate production (a proxy for the glycolytic rate), and the rate of oxygen consumption (OCR), as a measure of oxidative phosphorylation. Indeed, stimulation with IgA-IC alone increased the ECAR to a similar level as induced by TLR ligand Pam3CSK4. Importantly, this early glycolytic response was even further enhanced upon co-stimulation with IgA-IC and Pam3CSK4. In contrast, the OCR was not altered after any of these stimulations (Fig. 6a). We confirmed this increase in glycolysis upon FcαRI-TLR co-stimulation by measuring increased lactate accumulation in supernatants 24 h after stimulation (Fig. 6b). Next, to assess whether FcαRI-induced amplification of glycolysis is required for the induction of cytokine responses, we stimulated CD103[+] DCs in the presence of 2-deoxyglucose (2-DG), which blocks glycolysis by inhibiting hexokinase activity[41]. In contrast to conventional

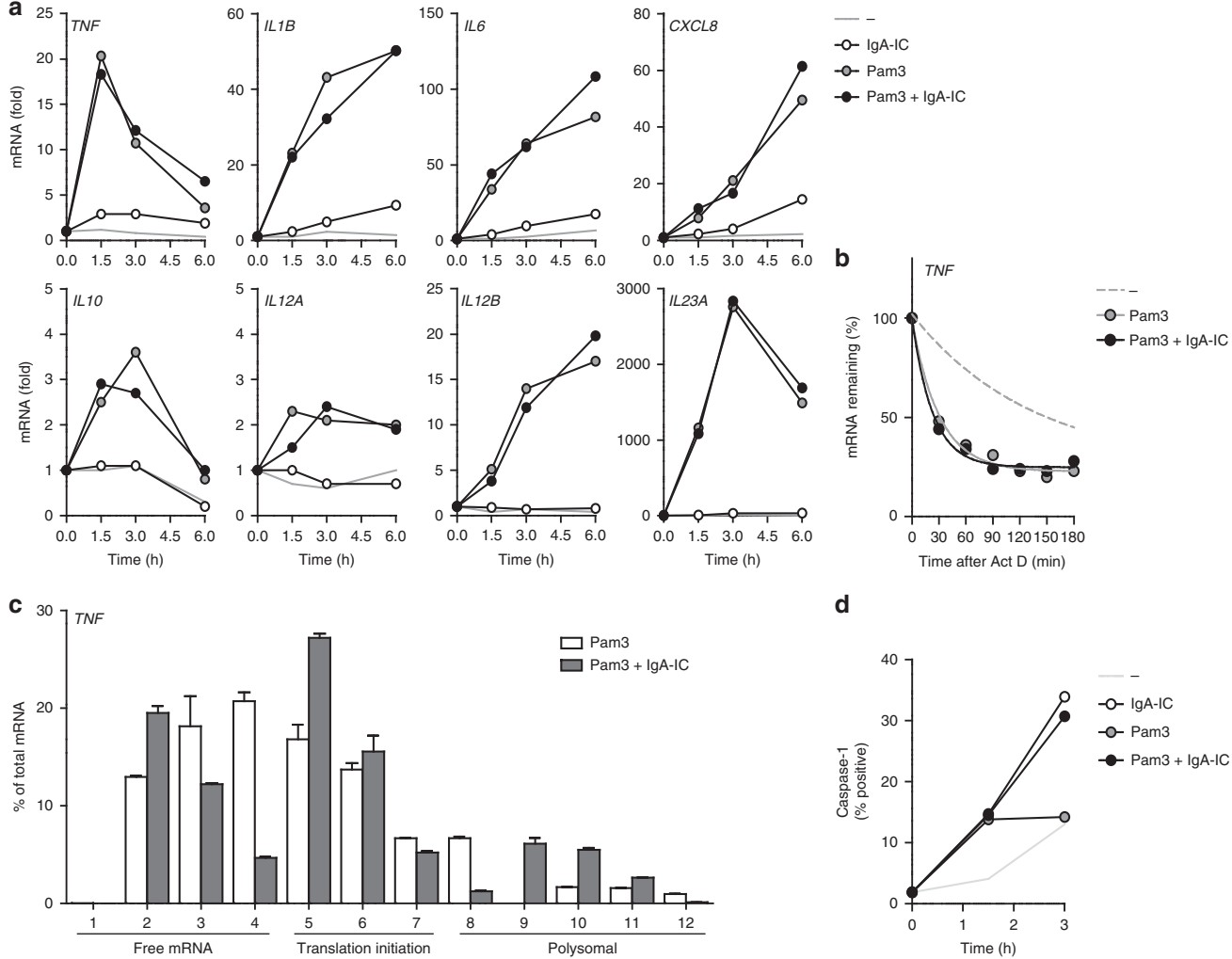

**Fig. 5** FcαRI stimulation promotes cytokine gene translation and induces caspase-1 activation. **a** CD103[+] dendritic cells (DC) were stimulated with Pam3CSK4 (Pam3), IgA immune complexes (IgA-IC), or a combination and analysed for mRNA expression of indicated genes using qPCR (normalized to *GAPDH* expression, fold increase compared to unstimulated control). **b** mRNA stability of *TNF* of CD103[+] DCs which were unstimulated or treated with Pam3CSK4 or co-stimulation with IgA-IC was determined after addition of 10 µg/mL actinomycin D (Act D) to prevent de novo synthesis of mRNA. *TNF* expression was analysed using qPCR (normalized to *GAPDH*). **c** Lysates of CD103[+] DCs stimulated for 3 h with Pam3CSK4 or Pam3CSK4 with IgA-IC were loaded on sucrose gradients to measure mRNA translation of *TNF* (normalized to *GAPDH*, qPCR was performed in duplicate, mean + s.e.m.). **d** Induction of caspase-1 activation in CD103[+] DCs after stimulation with Pam3CSK4, IgA-IC, or a combination was measured using caspase-1 binding compound FAM-YVAD-FMK (FLICA) by flow cytometry. Representative example (**a**–**d**) of three independent experiments

DCs[40,42], inhibition of glycolysis in CD103[+] DCs by 2-DG had little effect on proinflammatory cytokine production induced by TLR stimulation (Fig. 6c). However, strikingly, 2-DG substantially impaired cytokine amplification by FcαRI co-stimulation, indicating that in human CD103[+] DCs the enhanced glycolysis is specifically required for FcαRI-induced cytokine production. These data suggest that FcαRI increases glycolysis in CD103[+] DCs, which is essential for FcαRI-induced upregulation of cytokine gene translation.

Next, we aimed to identify the key components of FcαRI signalling responsible for the induction of glycolysis and subsequent production of proinflammatory cytokines by CD103[+] DCs. Several other FcαRI-mediated immune functions are induced by association of FcαRI with the FcR common γ-chain (FcRγ)[27], triggering activation of conserved upstream signalling pathways[27,43]. By screening inhibitors of FcαRI signalling molecules, we identified that inhibition of kinases Syk (by inhibitor R406 and si-RNA) and PI3K (by inhibitor wortmannin) specifically blocked FcαRI-induced production of proinflammatory cytokines by CD103[+] DCs, without affecting cytokine production induced by TLR stimulation alone (Fig. 6d, e, f). Overnight incubation of CD103[+] DCs with the inhibitors did not affect viability, indicating that the observed suppression was not the result of toxicity of the compounds (Supplementary Fig. 12). These data indicate that FcαRI-induced amplification of proinflammatory cytokines is dependent on kinases Syk and PI3K.

Subsequently, we set out to investigate how FcαRI signalling couples to increased glycolysis in CD103[+] DCs. In conventional murine DCs, amplification of glycolysis by TLR signalling is dependent on kinases TBK1 and IKKε[40,42]. However, similar to inhibition of glycolysis, in human CD103[+] DCs inhibition of TBK1-IKKε by inhibitors BX795 and Amlexanox blocked the amplification of cytokine production induced by IgA-IC (Fig. 6g, Supplementary Fig. 13). In line with these findings, inhibition of Syk, PI3K, and TBK1-IKKε impaired the IgA-IC-induced increase of ECAR without affecting the ECAR induced by Pam3CSK4 (Fig. 6h), indicating that these kinases are essential

for FcαRI-induced amplification of glycolysis. In addition, using

qPCR we verified that these key kinases are also expressed by

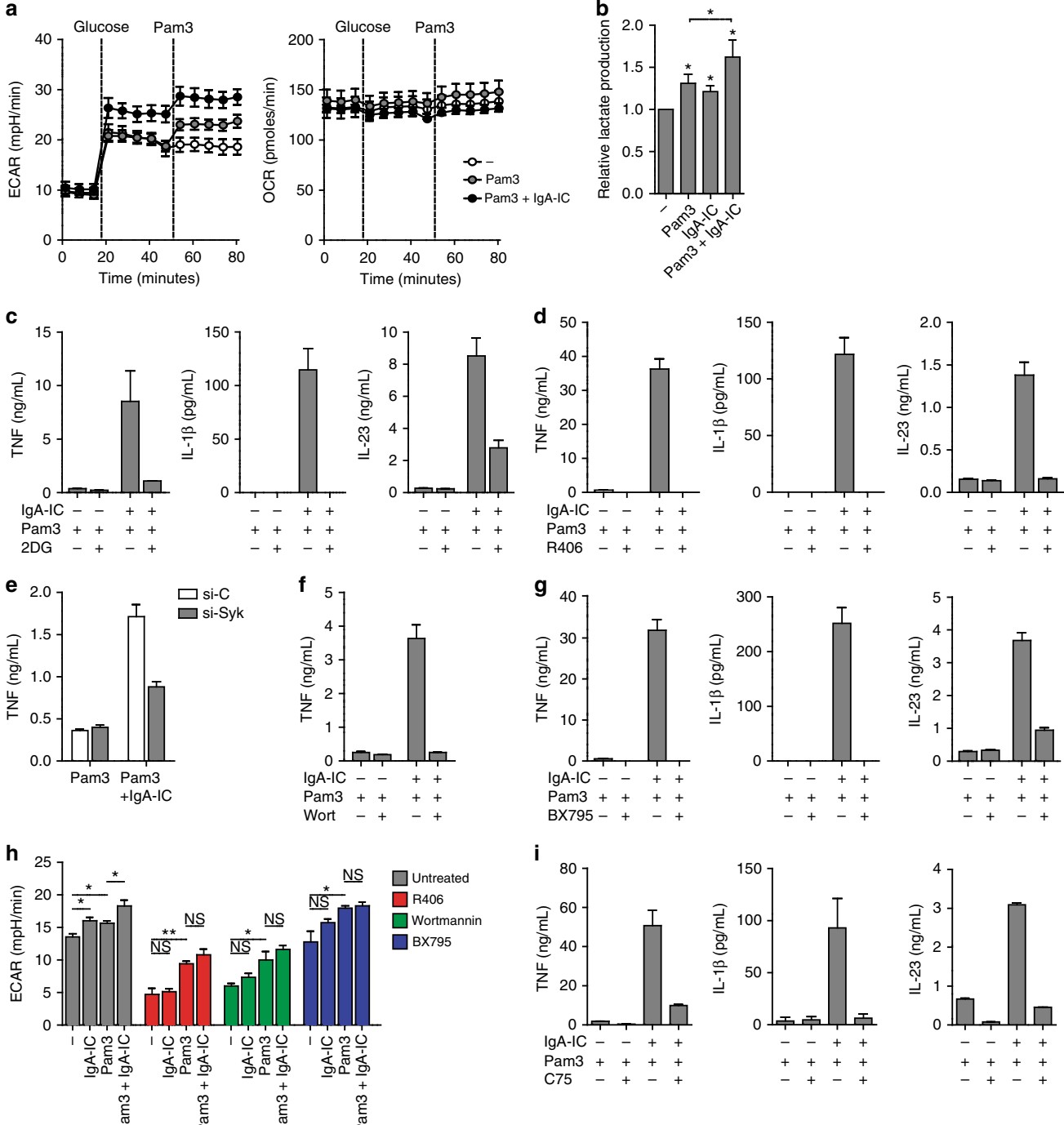

**Fig. 6** FcαRI-induced upregulation of cytokine production is dependent on glycolytic reprogramming through Syk, PI3K, and TBK1-IKKε. **a** Real-time changes in the extracellular acidification rate (ECAR) and oxygen consumption rate (OCR) of CD103[+] dendritic cells (DC) left unstimulated (open circles), stimulated with Pam3CSK4 (Pam3) alone (grey circles), or Pam3CSK4 with IgA immune complexes (IgA-IC) (black circles). IgA-IC are present from the start of the experiment. The dotted line indicates initiation of further treatment. Experiments were performed in quadruplicate, mean + s.e.m. **b** Relative lactate production of CD103[+] DCs after 24 h of treatment. Pooled data from five experiments. *$p < 0.05$. Student's $t$ test. **c**, **d** Cytokine production by CD103[+] DCs stimulated with Pam3CSK4, IgA-IC, or a combination after treatment with 10 mM glycolysis inhibitor 2-deoxy-D-glucose (2DG) (**c**) or 1 μM of syk inhibitor R406 (**d**). **e** TNF production by CD103[+] DCs silenced using specific si-RNA for Syk (si-Syk) or non-targeted control (si-C). **f**, **g** Cytokine production by CD103[+] DCs stimulated with Pam3CSK4, IgA-IC, or a combination after treatment with 100 nM PI3K inhibitor Wortmannin (Wort) (**f**) or 1 μM TBK1-IKKε inhibitor BX795 (**g**). **h** Differences in ECAR levels of CD103[+] DCs stimulated with Pam3CSK4, IgA-IC or a combination in the presence of inhibitors for Syk (R406), PI3K (Wort), or TBK1-IKKε (BX795). Experiments were performed in quadruplicate, mean + s.e.m. *$p < 0.05$, **$p < 0.01$, NS not significant. Student's $t$ test. **i** Cytokine production by CD103[+] DCs stimulated with Pam3CSK4, IgA-IC, or a combination after treatment with 20 μM fatty acid synthase inhibitor C75. Experiments **c**–**g**, **i** were performed in triplicate. After 24 h cytokine levels were analysed using ELISA, mean + s.e.m. Representative example (**a**, **c**–**i**) of three independent experiments

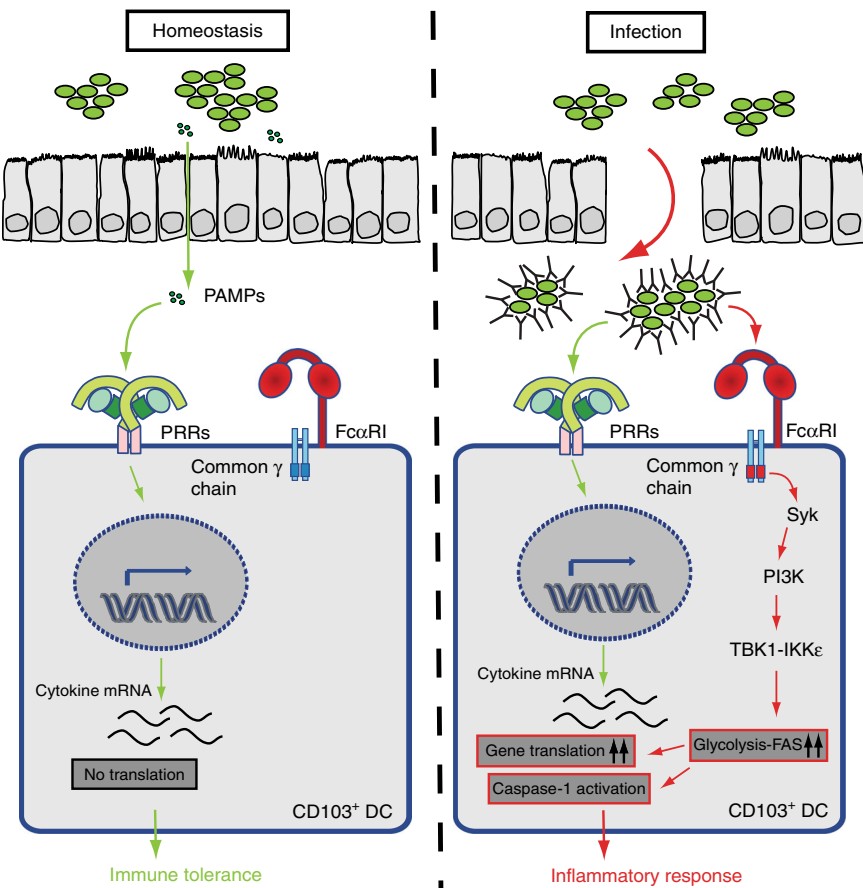

**Fig. 7** Model for inflammatory response upon FcαRI-TLR cross-talk. Under homeostatic conditions CD103[+] dendritic cells (DC) are activated by pathogen-associated molecular patterns (PAMP) through leakage or sampling of the lumen. This induces inflammatory cytokine mRNA transcription, but no protein production. Upon infection, IgA immune complexes (IgA-IC) provide a second signal through FcαRI triggering Syk, PI3K, and TBK1-IKKε activation, which increases the glycolytic flux and fatty acid synthesis (FAS) that results in an inflammatory response mediated by increased mRNA translation and caspase-1 activation

primary human intestinal CD103[+] DCs (Supplementary Fig. 14). Thus, in CD103[+] DCs FcαRI signalling couples to glycolytic reprogramming through kinases Syk, PI3K, and TBK1-IKKε.

DC activation and proinflammatory cytokine production has been shown to depend on increased fatty acid synthesis fuelled by glycolysis, resulting in expansion of endoplasmic reticulum and Golgi to support the increased demand for protein synthesis[42]. To assess the involvement of fatty acid synthesis in FcαRI-TLR cross-talk, synthesis of fatty acids was inhibited by fatty acid-synthase inhibitor C75. FcαRI-induced amplification of proinflammatory cytokine production was strongly inhibited by C75 (Fig. 6i), suggesting that the increased glycolysis induced by FcαRI is used to enhance fatty acid synthesis and promote the production of proteins such as proinflammatory cytokines.

Taken together, these data identify that FcαRI co-stimulation converts human CD103[+] DCs to secrete high levels of proinflammatory cytokines through Syk, PI3K, and TBK1-IKKε dependent glycolytic reprogramming.

## Discussion

In various tissues, recognition of pathogens by innate immune cells through PRRs is sufficient for the induction of proinflammatory responses[44]. However, in tissues characterized by high presence of commensal microorganisms such as the gastrointestinal tract, PRR activation is an ongoing phenomenon, and additional cues are required to discriminate between homeostatic conditions and infection. Here, we provide evidence

that in the human intestine the presence of IgA-IC in the lamina propria as a result of IgA opsonization of bacteria may act as a second 'danger' signal that converts the intestinal response from tolerogenic to proinflammatory through activation of human CD103[+]SIRPα[+] DCs. This proinflammatory response is characterized by selective amplification of TNF, IL-1β, and IL-23, which promotes both Th17 responses and ILC3 activation. Cytokine amplification is mediated by cross-talk of FcαRI with various PRRs by enhanced gene translation and caspase-1 activation, which critically depends on FcαRI-induced glycolytic reprogramming through signalling via Syk, PI3K, and TBK1-IKKε (for model see Fig. 7).

The prevailing concept regarding IgA in mucosal immunity is that it provides a non-inflammatory first line of defence through 'immune exclusion', i.e. binding of microorganisms to prevent them from penetrating the epithelial barrier[22,45]. In contrast, its potential to actively contribute to the initiation of inflammatory responses is less well known. This may be a consequence of the absence in (frequently used) mouse models of FcαRI, one of the main receptors responsible for regulating IgA-dependent inflammatory responses in humans[46]. Here, we have shown that IgA may orchestrate human intestinal inflammation through FcαRI-dependent activation of CD103[+] DCs by immune complexes. Although immune complexes are probably also present to some extent within the normal lamina propria due to influx of soluble antigens, this is likely very low in comparison to immune complex presence upon infection or other gastrointestinal

diseases[45]. Furthermore, our data are in agreement with previous findings that show that FcαRI stimulation with IgA-IC can induce inflammatory responses (reactive oxygen species production, degranulation, phagocytosis, etc.) by other cell-types, such as neutrophils, eosinophils, and monocytes[46]. In addition, our data show that FcαRI co-stimulation particularly increases cytokine production by CD103[+] DCs, but not by conventional moDCs. Notably, in the intestine only few FcαRI-positive cells are observed under homeostatic conditions, since intestinal macrophages lack FcαRI expression[47,48], and neutrophils and monocytes are mainly recruited after infection[49,50]. This may render CD103[+] DCs as one of the main cell types for the initial recognition of IgA-IC, underlining their importance for initiation of inflammation upon intestinal infection.

It has become clear that human intestinal CD103[+] DCs can be further divided into different subsets with distinct immunological functions[10,11]. Although we could not functionally compare different primary intestinal CD103[+] DC subsets because of scarcity of intestinal tissue and concomitant low yields, the phenotypical analysis of the in vitro CD103[+] DCs suggests that FcαRI-induced inflammation is particularly relevant for a subset of intestinal DCs that is characterized by expression of SIRPα[+]. CD103[+]SIRPα[+] DCs are the subset that is specialized in activation of Th17 and ILC3[11], which very nicely corroborates our findings that demonstrate that these cells (upon FcαRI co-stimulation) are able to selectively increase IL-17 production by T helper cells and IL-22 production by intestinal ILC3. Since amplification of cytokine production was more pronounced by in vitro CD103[+] DCs (resembling the SIRPα[+] subset) than by primary CD103[+] DCs (containing both SIRPα[+] and SIRPα[-] subsets), FcαRI-induced inflammation may be predominantly instigated by the CD103[+]SIRPα[+] subset of human intestinal DCs.

Cytokines have a crucial function in shaping the intestinal immune response[51]. We showed that FcαRI co-stimulation affected cytokine production by CD103[+] DCs in a selective manner, with particularly pronounced amplification of proinflammatory cytokines TNF, IL-1β, and IL-23. Importantly, these cytokines do not only perform a function in intestinal host defence by counteracting invading pathogens, but are also strongly associated with Crohn's disease (CD), one of the two main forms of inflammatory bowel disease (IBD)[52]. TNF exerts a variety of critical proinflammatory functions in the intestinal mucosa and plays a central role in IBD pathogenesis, which is evident by the current use of TNF inhibition as a standard therapy for IBD[51]. IL-1β and IL-23 are known to be crucial for intestinal immunity against extracellular pathogens by promoting both Th17 and ILC3 responses. In addition, these Th17-associated cytokines are implicated in the pathogenesis of CD[51], with anti-IL-23 antibodies as an important candidate for therapeutic use[53]. Indeed, our data showed that FcαRI co-stimulation of CD103[+] DCs promoted Th17 responses and induced production of hallmark-cytokine IL-22 by intestinal ILC3. While FcαRI co-stimulation of CD103[+] DCs promoted inflammation by enhancing IL-17 production by Th17 cells, intestinal ILC3 did not produce detectable levels of IL-17 but instead strongly upregulated IL-22 production. In the intestine, the main function of IL-22 is to activate epithelial cells to increase host resistance against pathogens, e.g. through mucus production, tight junction fortification, and production of bactericidal compounds[54], suggesting that FcαRI stimulation of CD103[+] DCs does not only induce inflammation, but also promotes tissue repair. Combined, these data indicate that FcαRI co-stimulation orchestrates a protective immune response to counteract intestinal infections, but may also suggest involvement in induction of inflammation in the context of CD.

We showed that FcαRI stimulation amplified cytokine production by human CD103[+] DCs in at least two different ways. First, FcαRI stimulation increased cytokine gene translation. Notably, FcαRI co-stimulation specifically increased the translation of particular genes such as TNF, but not of other genes such as IL1B. This is in line with previous findings in activated macrophages and DCs that loading of mRNA with ribosomes is gene-dependent, with particular upregulation of translation of genes such as TNF and IL23A, but not IL1B[55]. Although AU-rich elements (ARE) in the 3'untranslated region (3'UTR) of mRNA have been implicated in cytokine gene translation and are present in TNF and IL23A mRNA[56], the underlying mechanisms for these differences are still not completely clear, since AREs are also present in mRNA of cytokines that are not or only moderately affected, such as IL1B, IL6, and IL10[55,56]. Remarkably, while FcαRI upregulated cytokine production induced by a large variety of PRRs, no amplification was observed upon co-stimulation with TLR3. Notably, TLR3 stimulation alone, i.e. in absence of FcαRI stimulation, already induced very high cytokine protein levels. This may suggest that in CD103[+] DCs TLR3 signalling itself initiates a pathway to amplify gene translation, which is supported by previous findings that TLR3-induced TRIF signalling stimulates TNF gene translation in myeloid cells[57]. In addition to gene translation, a second mechanism of FcαRI stimulation activates caspase-1, which is required for cleavage of translated pro-IL-1β into its functional form[58]. As such, these data identified FcαRI as a second member of the Fc receptor family that can activate caspase-1, in addition to Fc gamma receptors (FcγR)[25]. Most likely, this FcαRI-induced caspase-1 activation is mediated by Syk-dependent amplification of glycolysis, similar to previously published data in mice that demonstrated that FcR-induced NLRP3 inflammasome activation is both Syk[59] and glycolysis dependent[60]. Interestingly, we showed that FcαRI-induced IL-1β production is also dependent on fatty acid synthesis. Since FcαRI co-stimulation does not affect IL1B gene translation, amplification of IL-1β production by fatty acid synthesis is most likely achieved through caspase-1 activation, which indeed has been demonstrated by others[61]. Combined, these data demonstrate that FcαRI co-stimulation elicits proinflammatory cytokine production by CD103[+] DCs through amplifying gene translation and activating caspase-1.

It has become clear that metabolic changes are key mediators in the orchestration of proinflammatory responses[39]. In immunogenic DCs, activation through one receptor (most notably a PRR) is sufficient for strong expression of proinflammatory cytokines, since this induces both proinflammatory gene transcription as well as orchestrates metabolic changes that allow for efficient translation of these genes[42]. In contrast, our data indicate that in tolerogenic CD103[+] DCs stimulation of only PRRs is insufficient for potent expression of proinflammatory cytokines, despite efficient induction of proinflammatory cytokine gene transcription and glycolysis by PRR stimulation. The reason for this lack of activation and consequently immunoregulatory phenotype of these cells is still unclear, but may be dependent on the activation of regulatory pathways such as noncanonical NF-κB signalling[62,63]. Here, we hypothesize that in CD103[+] DCs the PRR-induced changes in glycolysis are too low to enable efficient translation of these genes in these cells, resulting in a muted proinflammatory response. This may be an adaptation to their environment that is characterized by the continuous presence of microbial stimuli under homeostatic conditions. In this context, FcαRI appears to function as a 'safety switch' of CD103[+] DCs, in which only the combination of microbial components (activating PRRs) and IgA-IC (activating FcαRI), as occurs upon microbial invasion, is sufficient to push the glycolytic rate beyond a certain threshold to allow for sufficient synthesis of membranes required

for efficient translation of proinflammatory genes and full DC activation (Fig. 7 for model).

Our data identify FcαRI, a non-PRR, as a new receptor capable of inducing metabolic reprogramming. We showed that FcαRI promotes glycolysis in CD103[+] DCs through kinases Syk and PI3K, which couples to the common TBK1-IKKε-dependent pathway in DCs that enhances glycolysis to increase fatty acid synthesis and endoplasmic reticulum expansion to ultimately increase gene translation[42]. FcαRI signals by association with the FcR γ-chain (FcRγ) that bears an immunoreceptor tyrosine-based activation motif (ITAM), which belongs to the most widely used conserved signalling modules in leukocytes[64]. Interestingly, stimulation with IgG-IC, which activate FcγRs that also signal via ITAMs, upregulated proinflammatory cytokine production in a similar manner as IgA-IC. Although FcαRI stimulation is far more likely to occur than FcγR stimulation in the intestine due to the high relative expression of IgA versus IgG[65], these data suggest that induction of the identified pathway may also be induced by other ITAM-associated receptors, such as other Fc receptors, CLRs, NK receptors, and PSGL-1[66].

Taken together, these results indicate that formation of IgA-IC in the human intestine provides an environmental cue for the conversion of a tolerogenic to an inflammatory response by human CD103[+] DCs. While the physiological function of this phenomenon most likely is to provide protective immunity against invading microorganisms, the nature of the inflammatory response (characterized by specific amplification of TNF, IL-1β, IL-23, Th17, etc.) also suggests association with inflammation as observed in CD patients. In addition, FcαRI activation of CD103[+] DCs is also likely to occur in patients suffering from celiac disease, in which high concentrations of IgA-IC are found as a result of the presence of IgA antibodies against gluten as well as IgA autoantibodies against transglutaminase 2[67]. Hence, from a therapeutic point of view, targeting of FcαRI signalling or FcαRI-mediated metabolic changes may be a valuable approach to attenuate inflammation in chronic intestinal inflammatory disorders, including CD and celiac disease.

## Methods

**Dendritic cells**. All human tissues were collected after obtaining written informed consent in accordance with the approval of tissue-specific protocols by the Medical Ethical Committee of the Academic Medical Centre, Amsterdam. In vitro differentiated dendritic cells were obtained by isolation of monocytes from buffy coats (Sanquin Blood Supply) by density gradient centrifugation using Lymphoprep (Nycomed) and Percoll (Pharmacia). DCs were differentiated by culturing the monocytes for 6 days in IMDM (Lonza) containing 10% FBS (Biowest) and 86 μg/mL gentamycin (Gibco), supplemented with 20 ng/mL GM-CSF (Invitrogen), 2 ng/mL IL-4 (Miltenyi Biotec), and 1 μM retinoic acid (RA; Sigma Aldrich). At day 2 or 3 half of the medium was replaced with new medium containing cytokines.

BDCA1[+] DCs were isolated from buffy coats using MACS isolation with CD1c (BDCA1)[+] microbeads (Miltenyi Biotec) according to manufacturer's instructions.

Ex vivo CD103[+] DCs were obtained from surgically resected intestine. Single-cell suspension was acquired as described here[9]. In brief, lamina propria was separated from epithelial cells after incubation with PBS containing 5 mM EDTA for 30 min at 37 °C. Lamina propria was cut into small pieces and digested with RPMI (Gibco) containing 125 μg/mL Liberase TM for 30 min at 37 °C. Cell suspensions were filtered through a 70 μm nylon mesh. Cells were stained for CD45[+]CD11c[+]HLA-DR[+]CD103[+] and FACS-sorted (FACSaria, BD biosciences).

**Stimulation**. DCs were harvested at day 6 by removing medium and washing the cells with PBS and adding TrypLE select (Invitrogen). Cells (30,000–50,000 per well) were stimulated with 10 μg/mL Pam3CSK4 (Invivogen), 50 μg/mL LTA (Invivogen), 100 ng/mL LPS (from *E. coli* o111:B4; Sigma Aldrich), 20 μg/mL Poly I:C (Sigma Aldrich), 5 μg/mL CLO97 (Invivogen), 10 μg/mL PGN (*S. aureus*; Sigma Aldrich), 10 μg/mL MDP (Invivogen), 10 μg/mL curdlan (from *Alcaligenes faecalis*; Sigma Aldrich), and IgA-IC. *S. aureus* bacteria were opsonized by incubation with 5 mg/mL serum IgA, after which bacteria were washed to remove unbound IgA. Binding of serum IgA to bacteria was verified by a binding ELISA, as previously described[25]. In short, bacteria were coated overnight in PBS. Wells were washed and blocked using bovine serum albumin (Sigma Aldrich), subsequently incubated with 4 μg/mL IgA purified from human serum (MP biomedicals). Binding of IgA

was detected using anti-human IgA-HRP (Sanquin Blood Supply). Cells were stimulated with IgA opsonized bacteria in X-VIVO15 culture medium (Lonza) using 10 bacteria/DC. For IgA-IC stimulation, 96-well high-affinity Maxisorp plates (Nunc) were coated with 4 μg/mL serum IgA (MP biomedicals), purified IgA1 or IgA2 (both Alpha Diagnostics) diluted in PBS overnight at room temperature, followed by blocking with PBS containing 10% FBS for 1 h at 37 °C.

For silencing of Syk, CD103[+] DCs were harvested at day 3 using TrypLE select and microporated in the presence of 250 nM Syk SMARTpool si-RNA or control si-RNA (both Dharmacon) and cultured for three more days in the presence of GM-CSF, IL-4, and RA.

**ELISA**. Cytokine levels in supernatant were measured by ELISA, using antibody pairs for the following cytokines: IL-1β (CT213-c; CD2013-d), IL-6 (CD205-c; CD205-d), IL-13 (QS-13; LM-1), IL-23 (CD517-c; C8.6) IFN-γ (MD2; MD1; all U-CyTech), IL-10 (JES3-9D7; JES3-12G8; BD Pharmingen), TNF (MAb1; MAb11), and IL-17A (eBio64cap17; eBio4dec17; both eBioscience). IL-22 was measured using ELISA kit (88-7522-88, Thermo Fisher Scientific). Please note that absolute cytokine levels showed great donor-to-donor variation.

**Quantitative RT-PCR**. To determine mRNA levels cells were lysed at the indicated time points and afterwards mRNA extraction was performed using RNeasy Mini Kit (74106; Qiagen) and cDNA synthesis using RevertAid H Minus First Strand cDNA Synthesis Kit (K1632; Fermentas). Quantitative RT-PCR (StepOnePlus Real-Time PCR System; Thermo Fisher Scientific) was performed using Taqman Master Mix and the following Taqman primers (all from Thermo Fisher Scientific): *CXCL8* (Hs00174103_m1), *FCAR* (Hs02572026_s1), *GAPDH* (4310884E), *IKBKE* (Hs01063858_m1), *IL10* (Hs00961622_m1), *IL12A* (Hs01073447_m1), *IL12B* (Hs01011518_m1), *IL17A* (Hs00174383_m1), *IL17F* (Hs00369400_m1), *IL1B* (Hs00174097_m1), *IL22* (Hs01574151_m1), *IL23A* (Hs00372324_m1), *IL6* (Hs00174131_m1), *RORC* (Hs01076122_m1), *SYK* (Hs00895377_m1), *TBK1* (Hs00179410_m1), and *TNF* (Hs00174128_m1).

**Caspase-1 activation**. Caspase-1 activation was determined using caspase-1 binding compound FAM-YVAD-FMK from FAM-FLICA Caspase-1 Assay Kit (ICT098; ImmunoChemistry Technologies) according to manufacturer's instructions. Cells were assessed for fluorescence using flow cytometry (Canto II, BD Biosciences). For effect of caspase-1 inhibition on cytokine production cells were incubated with 20 μM Ac-YVAD-CMK (Invivogen) for 30 min at 37 °C and treated as indicated.

**Ribosome profiling**. For ribosome profiling of mRNA, cells were stimulated with Pam3CSK4 or in combination with IgA-IC for 3 h gently scraped and lysed using lysis buffer containing 20 mM Tris-HCl, 10 mM MgCl$_2$, 100 mM KCl, 1% Triton X-100, 2 mM DTT, EDTA-free complete protease inhibitor (Roche) and RNase inhibitor. Nuclei were removed using centrifugation at 1300 × g and lysates were loaded on a linear 7%–47% sucrose gradient and centrifuged for 2 h at 220,000 × g. Fractions were analysed for specific mRNAs using quantitative RT-PCR.

**Flow cytometry**. For phenotyping, CD103[+] DCs were stained using 5 μg/mL CD11c-FITC (3.9), CD141-FITC (JAA17), CD172a-APC (15–414) and CD370-PE (9A11; all eBioscience), HLA-DR-PE (G46-6), CD14-FITC (MφP9), CD80-FITC (L307.4), CD86-PE (2331; all BD Biosciences), CD1c-AF647 (L161), CD45-APC (2D1; Biolegend), CD83-APC (HB15e; Exbio), and CD103-PE (ber-act8; Sony) in PBS containing 0.5% BSA. FcαRI expression was determined by staining cells with 5 μg/mL anti-FcαRI (CD89; MIP8a; Abcam) followed by PE-conjugated goat-anti-mouse (Jackson Immunoresearch). Cells were analysed by flow cytometry (Canto II, BD Biosciences).

**FcαRI blockade and kinase inhibition**. FcαRI was blocked by incubating CD103[+] DCs with 20 μg/mL of anti-FcαRI for 30 min on ice. Subsequently stimuli and culture medium was added resulting in a final antibody concentration of 5 μg/mL. Kinases were inhibited by incubating CD103[+] DCs with selected small molecule inhibitors or equivalent amount of DMSO for 30 min at 37 °C. Inhibited kinases were Syk using 1 μM R406 (Selleckchem), PI3K using 100 nM wortmannin (Invivogen), and TBK1-IKKε using 1 μM BX795 (Invivogen) or 100 μM Amlex-anox (Abcam).

**Metabolism assays**. Lactate concentrations in the medium were determined using a NADH-based spectrophotometer assay as previously described[68]. Real-time analysis of the extracellular acidification rate (ECAR) and the oxygen consumption rate (OCR) of CD103[+] DCs were analysed using an XF-96 Extracellular Flux Analyzer (Seahorse Bioscience). XF-96 cell culture plates were coated with 8 μg/mL IgA. 30,000 cells were plated per well and assayed in glucose-free medium after addition of glucose at the indicated time cells were left unstimulated or stimulated with 10 μg/mL Pam3CSK4. For analysis with 2-Deoxy-D-glucose (2DG; Sigma Aldrich) or C75 (Sigma Aldrich) cells were incubated with 10 mM 2DG or 20 μM C75 for 30 min at 37 °C and treated as indicated for 24 h. Supernatant was stored at −20 °C and analysed by ELISA.

**Th polarization and ILC differentiation**. Memory CD4$^+$ T cells were isolated using MACS isolation (Miltenyi Biotec) with CD45RP-PE (Dako) and anti-PE beads (Miltenyi Biotec). 50,000 DCs were stimulated with Pam3CSK4 and /or IgA-IC and co-cultured with 50,000 allogeneic T cells in the presence of 10 pg/mL *S. aureus* enterotoxin B (SEB; Sigma Aldrich). After about 4 days cells were transferred to 96-well flat bottom culture plates (Greiner Bio-One). Every 2 days half of the medium was replaced by IMDM containing 10% FBS and 20 U/mL recombinant human IL-2 (Chiron). After ~12 days, resting T cells were re-stimulated with 1 µg/mL anti-CD3 (1XE; Sanquin Blood Supply) and 1 µg/mL anti-CD28 (15E8; Sanquin Blood Supply). Supernatant was harvested after 24 h and analysed by ELISA.

ILC3 were obtained from primary human tissue and expanded as described here[9]. In brief, Intestinal samples were processed as described earlier. Cells were stained with lineage markers (all FITC, CD1a (HI149), CD123 (6H6), BDCA2 (201 A), CD3 (okt-03), CD14 (HCD14), CD19 (HIB19), CD34 (581), TCRαβ (IP26), TCRγδ (B1), FcεR1α (AER-37), CRTH2 (BM16; all Biolegend), and CD16 (3G8; BD biosciences)), CD3-AF700 (UCHT1), CD45-APC (2D1), CD161-BV510 (HP-3G10), NKp44-AF647 (P44-8; all Biolegend), CD127-PEcy7 (R34.34), Ckit-PEcy5.5 (104D2D1; both Beckman Coulter), and CD94-PE (HP-3D9; BD Biosciences). ILC3 were sorted on CD45$^+$CD3$^-$Lin$^-$CD127$^+$CD94$^-$CD161$^+$Ckit$^+$NKp44$^+$ using FACSaria (BD biosciences). DCs were stimulated with Pam3CSK4 and/or c-IgA for 24 h afterwards supernatant was transferred onto the ILCs. The supernatant was supplemented with IL-2 and cultured for 7 days. Cells were lysed and mRNA was quantified using quantitative RT-PCR. Supernatant was analysed for IL-22 expression using ELISA.

**Statistical analysis**. Data was analysed for statistical significance using paired Student's *t*-test or Mann–Whitney test with GraphPad Prism version 5.01 software (GraphPad Software).

**Data availability**. All main data supporting the findings are available within the article or the Supplementary Information. The other data are available from the authors upon reasonable request.

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

## Acknowledgements

We thank Prof. Martien L. Kapsenberg for constructive criticism and Chrissta X. Maracle for critically reading the manuscript. This work was supported by grants from the Dutch Digestive Foundation (MLDS, Career Development Grant 2012); the Academic Medical Centre (AMC Fellowship 2015); and the Netherlands Organization for Scientific Research (NWO; VENI, project no. 91611012).

## Author contributions

Conceptualization, J.d.D.; Methodology, I.S.H., J.d.D., J.H.B., F.L.P., E.C.de.J., B.E., G.V., G.R.v.d.B.; Investigation, I.S.H., L.K., J.H.B., F.L.-P., W.H., J.A.v.B., E.C.K., B.E.; Resources, S.A.J.Z., C.J.B., W.A.B., M.E.W., R.A.; Writing—Original Draft, I.S.H., J.d.D.; Writing—Review & Editing, I.S.H., J.H.B., D.L.P.B., B.E., J.d.D.; Supervision, E.C.de.J., D. L.P.B., J.d.D.
