## [Peer Review File · Nature Communications]

Reviewers' comments:

Reviewer #1 (Mucosal immunity) (Remarks to the Author):

In this paper, Hansen et al describe the effects of IgA-containing immune complexes on human dendritic cells, presenting evidence that triggering of the FcαR can enhance TLR-mediated activation by promoting glycolysis-dependent cytokine production. The authors propose that this mechanism may be important during inflammatory conditions in the intestine, when IgA-bacteria complexes may gain access to CD103+ DC in the mucosa. This idea is novel and could help explain how normally "tolerogenic" DC might be activated in response to eg infection, where TLR triggering alone might not be sufficient. Most of the data presented substantiate the idea that these processes can occur in the in vitro models employed, but there are a number of issues which raise concerns over how relevant the results are to the intestine in vivo. Specific comments:

- 1) The work is based almost entirely on the use of an in vitro model that is claimed to generate "CD103+ DC". Even allowing for the fact that a simple culture system of this kind is unlikely to reproduce DC differentiation in a complex tissue such as the intestine, no information is provided on the phenotype of the cells that appear in vitro, or even whether they are homogeneous or not. Furthermore neither this study, nor the preceding ones using the model seem to have taken into account that it has been known for some time that the CD103+ population of intestinal DC is itself heterogeneous. As the subsets of CD103+ DC have quite distinct functions in vivo, can express different TLR and are differentially represented in other tissues, data using the total group of cells is impossible to interpret precisely. This is also important when considering the appropriate controls for such work, as the conventional monocyte-derived DC used here are likely to contain only one lineage of DC (cDC2) and may therefore not be representative of all the equivalent lineages found in eg spleen or other non-intestinal sites. This issue is compounded further by the recent understanding that such in vitro monocyte-derived "DC" may actually be more macrophage-like than genuine DC. For all these reasons, it is necessary to replicate more of the data using ex vivo derived intestinal DC rather than the single sample provided in a Supplementary figure.
- 2) A further fundamental issue is whether intestinal DC and their subsets express the FcαR1 in vivo. This is especially important given the very low levels found on the in vitro derived DC and it is important for these findings to be extended by more detailed Q-PCR and/or surface expression studies of the relevant DC in vitro and in vivo.
- 3) It seems unusual that the bacteria used with or without IgA opsonisation do not have more direct effects on the DC in vitro. In this respect, what are the levels of endotoxin in the IgA material itself?
- 4) The authors comment that the molecular pathways used by FcαR triggering overlap with those employed by other FcR. Therefore how specific are the effects of IgA-complexes? Can they be reproduced by eg IgG containing immune complexes?
- 5) Many of the results shown in the figures lack statistical analyses, meaning it is unclear whether the data shown are significantly different or that analysis has just not been performed.
- 6) In Figure 5C, only one sample appears to suggest that the IgA-complexes have a different effect on translation compared with the other conditions. In the absence of statistical analysis, is this sufficient to make conclusions?
- 7) The effects of the various inhibitors shown in Figure 6 are quite dramatic and given that these agents are known to have toxicity potential, it would be important to ensure that the treated cells are viable. This issue is underlined by the fact that cells stimulated with TLR alone produce no cytokines at all.
- 8) In this respect, R406 appears to markedly inhibit all responses, not just those in response to TLR + IgA-complexes; wortmannin has a similar, if lesser global effect.
- 9) Do the authors have any evidence that the uptake of IgA-bacteria complexes occurs in vivo as they propose?

Reviewer #2 (Immunometabolism) (Remarks to the Author):

Your manuscript, "Fc alpha receptor co-stimulation converts human intestinal CD103+ dendritic cells into proinflammatory cells through glycolytic reprogramming", shows a very interesting novel "gate-keeping" mechanism.

I do however think that a couple of additional experiments and clarifications in the manuscript are needed before I can recommend it for publication.

As this is a site-specific mechanism, the authors should show that this is really the case in intestinal CD103+ DCs in more than one sample. How did the other innate immune cell populations from this sort react to immunocomplex/ TLR stimulation (e.g. is it really specific to CD103+ DCs of the intestine or do other cell populations react similar)? How high is the expression of CD89 in various cells in such a sort?

Please be more precise in labeling your figures; it is not always clear from how many different donors/experiments the data is derived from. (Why is there no statistic most figures?)

I am not really convinced yet that ILC3s are mediating your inflammatory loop. In Fig 4 you show that co-cultivation of stimulated DCs and T cells induces them to secrete IL-17; and as this SN also promotes ILC3-Derived IL22 (in another experiment).

Why does rapamycin inhibit TNF production (partially) while Torin does not? In general you should complement your inhibitor data with some western blots, to show efficacy in your cell culture model.

"... inhibition of Syk and PI3K inhibited inhibited the IgA induced increase of ECAR without affecting Pam3C induced glycolysis ... (Fig 6H)" First of all I recommend to scale down the figure to what you actually want to show and put the rest in the supplements. What exactly do you want to compare here? Why does BX795 not affect the ECAR and does this inhibitor also block cytokine production?

Finally it would be very interesting to see if the translation-mediated effect is due to fatty acid synthesis/ ER-Golgi expansion.

Reviewer #3 (DC signalling, metabolism) (Remarks to the Author):

The bulk of previous studies on IgA indicated that the main function of these mucosal immune mediators is to induce immune exclusion, i.e., to provide a non-inflammatory line of defense. As a consequence of this, these Igs trap microorganisms in the intestinal lumen preventing their penetration to the lamina propria, where a dominant tolerogenic immune state is mainly maintained by CD103+ DCs. In fact, these DCs, at variance with 'classical' conventional DCs present in other tissues and organs, do not promote a pronounced inflammatory response when encounter microbial PAMPs, thus maintaining a homeostatic steady state in lamina propria. In the current work, Dennen and coworkers demonstrated that intestinal lamina propria CD103+ human DCs do promote a strong inflammatory immune response but only if co-stimulated by TLR ligands and IgA immune complexes (IgA-IC; such as IgA opsonized bacteria). This would imply that CD103+ DCs do activate an inflammatory, protective response when opsonized bacteria penetrate lamina propria. The Authors also demonstrated that the second danger signal (in addition to TLR activation) provided by IgA-IC relies not on transcriptional events but rather on translational mechanisms based on the activation of Syk, PI3K, and TTK1/IKKe, which increase glycolysis and consequently the synthesis of fatty acids (and thus expansion of endoplasmic reticulum to ultimately increase gene translation) in these cells.

This is a very nice work that opens a completely new perspective on the function of IgA and on the modalities of activation of CD103+ cells, findings that may also be important for IBDs where uncontrolled inflammation of lamina propria occurs. However, I have some minor concerns that would preclude the publication of the paper in its present form.

- 1) TLR signaling can also promote immunoregulatory rather than inflammatory effects in certain circumstances; thus the 'low grade' inflammatory response of CD103+ DCs may be due to the activation of such control pathways (i.e., noncanonical NF- κ B). The Authors should at least mention this possibility in the Discussion with the relevant references.
- 2) although published previously, the *S. aureus* model should be better explained here: for example, it is unclear here how IgA are selected in term of antigen specificity
- 3) at least in M&M, Authors should state that the absolute production of cytokines was very variable (in certain occasions TNF release changes from ng to pg/ml)
- 4) legend to Fig 1 is unclear, 'B' should be shifted closed to moDCs
- 5) Authors cannot state (in several figure panels, including Fig 1A, 2A but also several others) that a certain condition inhibits or stimulates something if statistical analysis is not shown
- 6) in Fig. 4B, I would suggest to extend the analysis also to IL-17F which has been shown to be more protective and less pathogenetic in mucosa; moreover, given the importance of IL-22 in gut protection, protein expression of this cytokine should also be shown
- 7) involvement of caspase-1 should also be proved by the use of a specific inhibitor of this enzyme; its concomitant activation does not demonstrate its involvement in the described mechanism
- 8) in the text commenting fig. S4, only lack of effect of torin is discussed. However, Fig. S4 also shows that rapamycin does inhibit. So, is or is not mTOR involved in the mechanism?
- 9) Fig S5: Authors stated in the main text that amlexanox did not affect TLR-induced cytokine production, which however is undetectable!
- 10) Fig 6A: although mentioned in the main text, the control with IgA-IC is missing
- 11) Fig. 6E: the panel is difficult to understand because si-Syk abrogates completely TNF-alpha production and should be made clearer

We thank the reviewers for their comments and suggestions and are glad to hear from the editor that they consider this work of considerable potential interest for publication in Nature Communications. The revision of this manuscript took a considerable amount of time because of the extreme scarcity of the required human intestinal tissue, from which multiple cell types (dendritic cells subsets, as well as innate lymphoid cells) needed to be isolated. However, we now believe that we have been able to address all questions of the reviewers.

Since some questions were raised by multiple reviewers (primary DCs, statistics, rapamycin), we first address these questions. Subsequently, we have made a point-by-point response to the individual questions of the reviewers.

Primary DCs

Reviewer #1: *1) The work is based almost entirely on the use of an in vitro model that is claimed to generate "CD103+ DC". Even allowing for the fact that a simple culture system of this kind is unlikely to reproduce DC differentiation in a complex tissue such as the intestine, no information is provided on the phenotype of the cells that appear in vitro, or even whether they are homogeneous or not. Furthermore neither this study, nor the preceding ones using the model seem to have taken into account that it has been known for some time that the CD103+ population of intestinal DC is itself heterogeneous. As the subsets of CD103+ DC have quite distinct functions in vivo, can express different TLR and are differentially represented in other tissues, data using the total group of cells is impossible to interpret precisely. This is also important when considering the appropriate controls for such work, as the conventional monocyte-derived DC used here are likely to contain only one lineage of DC (cDC2) and may therefore not be representative of all the equivalent lineages found in eg spleen or other non-intestinal sites. This issue is compounded further by the recent understanding that such in vitro monocyte-derived "DC" may actually be more macrophage-like than genuine DC. For all these reasons, it is necessary to replicate more of the data using ex vivo derived intestinal DC rather than the single sample provided in a Supplementary figure.*

Reviewer #2: *As this is a site-specific mechanism, the authors should show that this is really the case in intestinal CD103+ DCs in more than one sample.*

Even though the used in vitro model resembles intestinal CD103⁺ DCs (phenotype, homogeneity, and function are extensively described in ¹), the reviewers are absolutely right that the in vitro generated cells are unlikely to fully resemble primary intestinal cells. Therefore, as requested by the reviewers, we have tested the response of primary CD103⁺ DCs isolated from human intestine for additional donors (sorting strategy is depicted in Suppl. Figure 2). Importantly, as shown in Figure 2D, primary human CD103⁺ DCs isolated from the intestine of multiple donors indeed amplified pro-inflammatory cytokine production upon co-stimulation of TLR ligands with complexed IgA, similar to the in vitro model. Although further subdivision into different subsets of CD103⁺ DCs was unfortunately impossible due to the limited amount of intestinal tissue per donor and the very low yield after cell sorting, these data confirm that the intestinal CD103⁺ DC population as a whole indeed promotes inflammation upon combined stimulation with TLR ligands and IgA immune complexes.

Reviewer #1: *2) A further fundamental issue is whether intestinal DC and their subsets express the FcαR1 in vivo. This is especially important given the very low levels found on the in*

in vitro derived DC and it is important for these findings to be extended by more detailed Q-PCR and/or surface expression studies of the relevant DC *in vitro* and *in vivo*.

As requested, we have extended the analysis of Fc α RI expression to different subsets of primary DCs. As shown in Supplementary Figure 4, non-intestinal DCs such as BDCA1⁺ DCs isolated from blood showed very low levels of Fc α RI expression, which is in line with the common conception that Fc α RI is not expressed by human DCs². In stark contrast, Fc α RI expression is substantially higher in CD103⁺ DCs isolated from human intestine (Supplementary Figure 4). These data further illustrate that intestinal CD103⁺ DCs, in contrast to DC subsets in other regions of the body, have a phenotype and function that is specialized to tailor responses to the intricate immunological conditions in the intestine.

Reviewer #2: *How did the other innate immune cell populations from this sort react to immunocomplex/ TLR stimulation (e.g. is it really specific to CD103⁺ DCs of the intestine or do other dell populations react similar)? How high is the expression of CD89 in various cells in such a sort?*

Due to the very low yield after cell sorting it was difficult to extensively test the response of other primary intestinal DC subsets. However, from two donors we obtained a sufficient amount of CD103⁻ DCs to test the effect of Fc α RI-TLR co-stimulation. As expected, primary intestinal CD103⁻ DCs cells expressed less Fc α RI than CD103⁺ DCs (Figure for reviewers 1A). However, also CD103⁻ DCs (moderately) increased TNF α production upon Fc α RI co-stimulation (Figure for reviewers 1B). Nevertheless, the relative increase in cytokine production (i.e. (Fc α RI+TLR) / TLR) appears to be higher for primary CD103⁺ DCs compared to primary CD103⁻ DCs, since individual TLR stimulation of CD103⁺ DCs is often very low/undetectable (Figure 2D). Although these preliminary data on additional human intestinal DC subsets is certainly interesting for future research, we have not included this in the manuscript but made a separate figure for the reviewer, since we feel that the scope of this paper is the inflammatory function of the CD103⁺ DC subset.

Figure for Reviewers 1.

A mRNA expression of Fc α RI-encoding gene *FCAR* (normalized to *GAPDH*) in BDCA1⁺ DCs from blood, and CD103⁻ and CD103⁺ DCs isolated from human intestine. One representative sample. **B** TNF α protein production by CD103⁻ DCs isolated from human intestine after stimulation with Pam3CSK4 or Pam3CSK4 combined with IgA-IC. Each pair of dots represents one donor.

Statistics

Reviewer #1: *5) Many of the results shown in the figures lack statistical analyses, meaning it is unclear whether the data shown are significantly different or that analysis has just not been performed.*

Reviewer #2: *Please be more precise in labeling your figures; it is not always clear from how many different donors/experiments the data is derived from. (Why is there no statistic most figures?)*

Reviewer #3: *5) Authors cannot state (in several figure panels, including Fig 1A, 2A but also several others) that a certain condition inhibits or stimulates something if statistical analysis is not shown*

We can understand the confusion, since in the original manuscript the figure legends indeed did not always state that a depicted figure was a representative example. In other cases, figure legends sometimes did not mention how many donors were tested. As requested, we have adjusted the figure legends, which now all clearly state whether a representative example is shown, as well as the number of individual donors that were tested for each experiment.

Please note that, for general clarity of the paper, in this manuscript for most experiments we show representative examples, since absolute levels of cytokine production show very large donor-to-donor differences (as also acknowledged by Reviewer #3 (point 3)). While for a representative example of a single donor (even though tested in triplicate) statistical calculations are not commonly applied, for key findings we have included data for multiple donors for which we have calculated statistical differences. For example, Figure 2A and 2B show the same data: Figure 2A is a representative example (statistics not applied), while Figure 2B shows multiple donors (statistics applied, with significant differences in all conditions). Similarly, statistics have been applied for multiple donor experiments such as Figure 6B and Figure S1.

Rapamycin

Reviewer #2: *Why does rapamycin inhibit TNF production (partially) while Torin does not? In general you should complement your inhibitor data with some western blots, to show efficacy in your cell culture model.*

Reviewer #3: *8) in the text commenting fig. S4, only lack of effect of torin is discussed. However, Fig. S4 also shows that rapamycin does inhibit. So, is or is not mTOR involved in the mechanism?*

We were also somewhat surprised to see that rapamycin (which inhibits mTORC1) induced a partial inhibition of TNF α , while Torin (which inhibits both mTORC1 and mTORC2) did not have any effect. First, we investigated whether this suppression is caused by toxicity, but rapamycin was not toxic for the CD103⁺ DCs in the used concentration (Suppl. Figure 8).

Second, we took a closer look at the effect of rapamycin on TLR stimulation versus Fc α RI-TLR co-stimulation. As shown in Suppl. Figure 7B, rapamycin did not only partially block TNF α production after Fc α RI-TLR co-stimulation, but also blocked TNF α production induced by (individual) TLR stimulation. However, as shown for multiple donors in Figure for Reviewers 2A, Fc α RI co-stimulation still clearly amplified pro-inflammatory cytokine production in the presence of rapamycin. In fact, the relative increase in TNF α production by Fc α RI co-stimulation is similar in the presence of rapamycin

compared to the control situation (Suppl. Figure 7C). These data indicate that rapamycin mainly affects cytokine production induced by TLRs, but has relatively little effect on cytokine production induced by co-stimulation with Fc α RI. When combining these data with the finding that Torin did not have any effect on TNF α production (neither induced by Fc α RI-TLR, nor by individual TLR stimulation) (Suppl. Figure 7A), we suspect that the partial TNF α suppression by rapamycin is induced by non-specific (off-target) effects of this small molecule inhibitor on TLR activation.

Figure for Reviewers 2

TNF α production by CD103⁺ DCs upon stimulation with Pam3CSK4 or Pam3CSK4 combined with IgA-IC in the presence of DMSO or Rapamycin. Each pair of dots represents one donor. *P <0.05, **P <0.01, ***P <0.001, Mann Whitney test.

Individual questions of reviewers

Reviewer #1

3) It seems unusual that the bacteria used with or without IgA opsonisation do not have more direct effects on the DC in vitro. In this respect, what are the levels of endotoxin in the IgA material itself?

We believe that here the reviewer underlines an important feature of CD103⁺ DCs compared to conventional DCs. Conventional DCs produce high levels of pro-inflammatory cytokines in response to (un-opsonized) bacteria (Figure 1B). In contrast, CD103⁺ DCs show very little response to un-opsonized bacteria, but do amplify pro-inflammatory cytokine production upon IgA opsonization of bacteria (Figure 1A). We believe that this difference most likely reflects the tolerogenic function of CD103⁺ DCs under homeostatic conditions, during which an inflammatory response to commensals and their microbial components is undesirable. Only during infection, when bacteria become opsonized by high levels of IgA in the lamina propria, CD103⁺ DCs will promote inflammation. As an additional note, we would like to stress that the absolute levels of cytokine production by an individual donor should be interpreted with caution, since absolute levels show large (more than 10 fold) donor-dependent variations.

Although endotoxin levels of IgA are most likely extremely low, the manufacturer (MP Biomedicals) could not provide us with exact numbers. To determine any functional inflammatory effect of potential endotoxin in the IgA, we incubated conventional (monocyte-derived) DCs with IgA-IC and measured cytokine production. While conventional DCs (in contrast to CD103⁺ DCs) are highly

sensitive to endotoxin, IgA stimulation did not have any effect on cytokine production by these cells (Figure for reviewer 3). These data demonstrate that even if IgA would contain trace amounts of endotoxin, these levels are almost certainly too low to affect cytokine production.

Figure for Reviewers 3

moDCs were stimulated with IgA-IC or Pam3CSK4. Experiments were performed in triplicate. After 24h cytokine levels were analyzed using ELISA, mean + SEM. Representative example of five experiments using different donors.

4) *The authors comment that the molecular pathways used by FcαR triggering overlap with those employed by other FcR. Therefore how specific are the effects of IgA-complexes? Can they be reproduced by eg IgG containing immune complexes?*

As requested, we have also examined the effect of IgG immune complexes on CD103⁺ DCs. As shown in new Figure S3, IgG-IC also display cross-talk with TLRs, indicating that this is not restricted to FcαRI and may indeed be a more general function of ITAM signaling receptors. However, given that in the intestine IgA is expressed to a far greater extent than IgG³, the effect of IgG-IC on intestinal CD103⁺ DCs is most likely very limited (with the possible exception of individuals that are IgA deficient, in which IgG may actually compensate for the lack of FcαRI-TLR cross-talk).

6) *In Figure 5C, only one sample appears to suggest that the IgA-complexes have a different effect on translation compared with the other conditions. In the absence of statistical analysis, is this sufficient to make conclusions?*

In fact, Figure 5C indicates a shift in translation of *TNFA* in multiple ways. First, from fraction 4 to fraction 5, which demonstrates a transition from free mRNA (no translation at all) to translation initiation. Second, from fraction 8 to fractions 9, 10, and 11, which demonstrates a strongly increased binding of multiple ribosomes per mRNA of *TNFA*. Since in general polysomes are responsible for the majority of gene translation, the up-regulation of pro-inflammatory cytokines by FcαRI co-stimulation is most likely induced by the shift in these polysomal fractions.

7) *The effects of the various inhibitors shown in Figure 6 are quite dramatic and given that these agents are known to have toxicity potential, it would be important to ensure that the treated cells are viable. This issue is underlined by the fact that cells stimulated with TLR alone produce no cytokines at all.*

We agree that, when working with small molecule inhibitors, there is always a potential risk of toxicity. Therefore, we examined the toxicity of all inhibitors used in this study. As shown in Suppl. Figure 8, CD103⁺ DCs were still viable after 24h incubation with all tested inhibitors, indicating that the observed suppression of cytokine production indeed results from specific inhibition of FcαRI-induced signaling.

8) In this respect, R406 appears to markedly inhibit all responses, not just those in response to TLR + IgA-complexes; wortmannin has a similar, if lesser global effect.

Since CD103⁺ DCs are tolerogenic cells that induce little cytokine production in response to (individual) TLR stimulation, the effect of inhibitors or silencing of signaling molecules on TLR stimulation (in absence of IgA immune complexes) is often difficult to assess. To illustrate that inhibition of the FcαRI-induced signaling molecules does not affect individual TLR stimulation, we have replaced Figure 6E with data of a donor that did induce detectable levels of TNFα after individual TLR stimulation. Importantly, Syk silencing of the CD103⁺ DCs of this donor selectively suppressed TNFα production induced by FcαRI-TLR cross-talk, while it did not have any effect on cytokine production induced by TLR stimulation alone (Figure 6E). These data, combined with inhibitor data in Figure 6C, D, F, and G (in which very little effect of inhibitors is observed on cytokine production induced by individual TLR stimulation) demonstrate that specifically FcαRI-TLR cross-talk (and not individual TLR signaling) is dependent on kinases Syk, PI3K, and TBK1/IKKε in these cells.

9) Do the authors have any evidence that the uptake of IgA-bacteria complexes occurs in vivo as they propose?

Considering the extensive amount of previous reports involving various cell types that demonstrate that FcαRI induces phagocytosis of IgA-opsonized particles, coupled to the phagocytic capacity of DCs in general, we consider it to be very likely that CD103⁺ DCs take up IgA opsonized bacteria. However, it is important to realize that our data indicate that phagocytosis is not essentially required for FcαRI-TLR cross-talk, since stimulation with IgA immune complexes by using immobilized IgA that is coated on a plate (and therefore cannot be internalized by the cell) is very well capable of amplifying pro-inflammatory cytokine production by CD103⁺ DCs.

Reviewer #2

I am not really convinced yet that ILC3s are mediating your inflammatory loop. In Fig 4 you show that co-cultivation of stimulated DCs and T cells induces them to secrete IL-17; and as this SN also promotes ILC3-Derived IL22 (in another experiment).

We agree with the reviewer that IL-22 production by ILC3s, in contrast to IL-17 production by T cells, does not predominantly mediate inflammation. In the intestine, the main function of IL-22 is to activate epithelial cells to increase host resistance against pathogens, e.g. through mucus production, tight junction fortification, and production of bactericidal compounds⁴. Since FcαRI-TLR co-stimulation of CD103⁺ DCs clearly amplifies IL-22 production by intestinal ILC3s (we now also show this on protein level in new Figure 4C), infection of the lamina propria by pathogens most likely does not only activate CD103⁺ DCs to promote inflammation through pro-inflammatory cytokine

production (such as TNF α) and Th17 responses, but simultaneously also promotes tissue repair by the intestinal epithelium. We have clarified this in the Discussion of the manuscript (p. 15, line 333).

“... inhibition of Syk and PI3K inhibited inhibited the IgA induced increase of ECAR without affecting Pam3C induced glycolysis ... (Fig 6H)” First of all I recommend to scale down the figure to what you actually want to show and put the rest in the supplements. What exactly do you want to compare here? Why does BX795 not affect the ECAR and does this inhibitor also block cytokine production?

We can imagine the confusion, since (1) in the original version the text of the results section did not chronologically discuss the data in Figure 6, and (2) the original panel of Figure 6H was not clearly structured. First, we have adjusted the text of the results section, which now discusses the results in a chronological order. Second, we have divided the original Figure 6H into 4 separate panels to emphasize the relevant differences (new Figure 6H). Most important in Figure 6H is that (in absence of inhibitors) Fc α RI (co-)stimulation increases the ECAR, while inhibitors of Syk, PI3K, and TBK1/IKK ϵ counteract the Fc α RI-induced ECAR. Please note that TBK1/IKK ϵ inhibitor BX795 blocks both Fc α RI-induced ECAR (Figure 6H, most right panel) as well as cytokine production (Figure 6G).

Finally it would be very interesting to see if the translation-mediated effect is due to fatty acid synthesis/ ER-Golgi expansion.

As suggested by the reviewer, we have assessed the involvement of fatty acid synthesis by treatment of CD103⁺ DCs with fatty acid synthase inhibitor C75. As shown in Figure 6I, C75 inhibited Fc α RI-induced production of pro-inflammatory cytokines, indicating that the amplification of gene translation is indeed mediated by increased fatty acid synthesis. We thank the reviewer for this interesting suggestion.

Reviewer #3

1) TLR signaling can also promote immunoregulatory rather than inflammatory effects in certain circumstances; thus the ‘low grade’ inflammatory response of CD103+ DCs may be due to the activation of such control pathways (i.e., noncanonical NF- κ B). The Authors should at least mention this possibility in the Discussion with the relevant references.

We agree with the reviewer that the underlying mechanism that is responsible for the immunoregulatory phenotype of CD103⁺ DCs is still largely unclear, and that control pathways such as non-canonical NF- κ B may play a role in this. As requested, we have added this to the Discussion (p. 17, line 371).

2) although published previously, the S. aureus model should be better explained here: for example, it is unclear here how IgA are selected in term of antigen specificity

As requested, in addition to referring to previous publications, we have now more extensively explained the *S. aureus* IgA opsonization model in the Materials and Methods section (p. 19, line 427). For opsonization we incubated *S. aureus* bacteria in 5 mg/mL pooled (multiple donors) human serum IgA, after which bacteria were washed to remove non-specific IgA. Binding of IgA to *S. aureus* was verified using a binding ELISA, thereby confirming that bacteria were indeed IgA opsonized.

3) at least in M&M, Authors should state that the absolute production of cytokines was very variable (in certain occasions TNF release changes from ng to pg/ml)

We agree with the reviewer that, to avoid confusion, it is good to mention that the (absolute) levels of cytokine production shows large donor-dependent variation (see also point 3 of Reviewer 1). Therefore, as requested, we have added this to the Materials and Methods section (p. 20, line 447).

4) legend to Fig 1 is unclear, 'B' should be shifted closed to moDCs

As requested, the legend of Figure 1 has been adjusted.

6) in Fig. 4B, I would suggest to extend the analysis also to IL-17F which has been shown to be more protective and less pathogenetic in mucosa; moreover, given the importance of IL-22 in gut protection, protein expression of this cytokine should also be shown

As requested, we have measured IL-17F expression by human intestinal ILCs. However, as shown in updated Figure 4B, IL-17F expression was undetectable. Nevertheless, we thank the reviewer for this relevant suggestion. In addition, we have now also determined IL-22 production by ILCs at protein level (new Figure 4C), which showed that IL-22 up-regulation was very similar to mRNA level (Figure 4B).

7) involvement of caspase-1 should also be proved by the use of a specific inhibitor of this enzyme; its concomitant activation does not demonstrate its involvement in the described mechanism

To verify that caspase-1 is involved in Fc α RI-TLR cross-talk, we tested the effect of caspase-1 inhibitor Ac-YVAD-CMK on cytokine production by CD103⁺ DCs. As shown in new Figure S6, blocking of caspase-1 indeed resulted in specific inhibition of IL-1 β protein production, without affecting TNF α production.

9) Fig S5: Authors stated in the main text that amlexanox did not affect TLR-induced cytokine production, which however is undetectable!

Since TLR-induced production of TNF α is often undetectable, the effect of amlexanox on (individual) TLR stimulation is indeed difficult to assess. Therefore, we have removed the statement that amlexanox does not affect TLR-induced cytokine production. However, please note that BX795 (the other TBK1/IKK ϵ inhibitor that we tested) had very little effect on TLR-induced cytokine production (Figure 6G, most clearly for IL-23), indicating that in CD103⁺ DCs TBK1/IKK ϵ is not involved TLR-induced cytokine production.

10) Fig 6A: although mentioned in the main text, the control with IgA-IC is missing

Please note that in Figure 6A the ECAR is measured in real-time. When glucose is added (at 20 min.), cells are either unstimulated (indicated in white and grey) or stimulated with IgA-IC alone (indicated by black). Only after 50 min. Pam3 was added to the wells, which then shows cells that are unstimulated (in white), Pam3 stimulated (in grey), and Pam3 + IgA-IC (in black). Long story short: IgA-IC stimulation alone are the black dots between 20 and 50 minutes.

11) Fig. 6E: the panel is difficult to understand because si-Syk abrogates completely TNF-alpha production and should be made clearer

We agree that the effect of si-Syk on TNF α production is difficult to assess because of the complete inhibition and the lack of cytokine induction by individual TLR stimulation. Therefore, we have replaced Figure 6E with data of a donor that did induce detectable levels of TNF α after individual TLR stimulation. Importantly, Syk silencing of the CD103⁺ DCs of this donor selectively suppressed TNF α production induced by Fc α RI-TLR cross-talk, while it did not have any effect on cytokine production induced by TLR stimulation alone (new Figure 6E). These data demonstrate that Syk is specifically required for TNF α induction by Fc α RI-TLR cross-talk, but not by individual TLR stimulation.

A final remark: for clarity, we have moved the graphic summary of this manuscript from the supplementary to the main figures (new Figure 7).

Combined, we believe that the additional experimental data and textual adjustments have greatly strengthened the manuscript. We thank the reviewers for their comments and suggestions and look forward to their reply.

Kind regards,

Jeroen den Dunnen (also on behalf of all co-authors)

Reference List

1. Bakdash G, Vogelpoel LT, van Capel TM, Kapsenberg ML, de Jong EC. Retinoic acid primes human dendritic cells to induce gut-homing, IL-10-producing regulatory T cells. *Mucosal Immunol* **8**, 265-278 (2015).
2. Bakema JE, van Egmond M. The human immunoglobulin A Fc receptor Fc α RI: a multifaceted regulator of mucosal immunity. *Mucosal Immunol* **4**, 612-624 (2011).
3. Bjerke K, Brandtzaeg P, Rognum TO. Distribution of immunoglobulin producing cells is different in normal human appendix and colon mucosa. *Gut* **27**, 667-674 (1986).
4. Schreiber F, Arasteh JM, Lawley TD. Pathogen Resistance Mediated by IL-22 Signaling at the Epithelial-Microbiota Interface. *J Mol Biol* **427**, 3676-3682 (2015).

Reviewers' comments:

Reviewer #1 (Remarks to the Author):

While the authors have added material to this manuscript and addressed some of the original comments, some major issues remain unresolved. As a result of these, it is still unclear whether the FcαR-mediated phenomenon and its signaling pathways are a specific function of bona fide intestinal DC in vivo and if so, if this is a property of a defined population of these cells:

1) Although I fully appreciate the difficulties in examining individual subsets of CD103+ DC from human intestine, the authors continue to ignore the fact that such subsets exist in the intestine and these are radically distinct in phenotype, function and origin. There is now a substantial literature on these cells and it is not acceptable simply to group "CD103+" DC together as an entity for the kind of work presented here.

2) In this respect, the authors still do not provide any phenotypic information on the DC populations they derive in vitro. As a result, the reader does not know how many of these cells are genuine DC, let alone what proportion are CD103+ or which subset they might belong to.

3) Despite now providing data on primary intestinal DC in Figure 2D, the amounts of TNFα produced here are very small, no statistics are provided and it is not stated where the "intestinal" samples were obtained from. This last point is important, as there are clear anatomical differences between the subsets of DC found in distinct segments of the large and small intestine.

4) The expression of FcαR1 on intestinal (from where?) CD103+ DC shown in Figure 3a is very low, while the PCR results shown in Figure S4 appear to represent n=1. Given the previous view that DC do not generally express this receptor, more extensive analysis of protein and mRNA by appropriately identified DC subsets and other myeloid cells is needed to allow a full appreciation of the data presented here.

5) Several figures still lack statistic analysis and the provision of replicate numbers now reveals how few these often are. Of particular note, the data shown in Figure 5C appear to represent n=2, which means that it is invalid to calculate errors and makes the very small differences difficult to interpret.

6) In this respect, the data in Figure 5D appear to reflect n=1.

7) No studies of signaling molecule expression or function in DC in vivo have been carried out.

8) The new data on inhibition of kinase signaling shown in Figure 6H do not show large differences, lack statistical analysis and different scales are used on the separate panels, compromising interpretation.

9) How do the authors explain the new findings presented in Figure 6I, which appear to show parallel effects on all parameters assessed, contradicting the earlier findings that TNFα and IL1 may be regulated independently?

Reviewer #2 (Remarks to the Author):

The authors have responded well to my concerns. I do not have any further comments.

Reviewer #3 (Remarks to the Author):

All the concerns raised by this reviewer have been addressed in a satisfactory fashion

Dear Reviewer #1, dear editor,

We are very excited to hear that two out of three reviewers (Reviewer #2 and Reviewer #3) were completely satisfied with our response. Reviewer #1 had a number of remaining and additional questions and remarks, which we will address below.

Since the editor indicated Point 2 and Point 4 to be most important, we address these points first. Subsequently, we have made a point-by-point response to the other questions of Reviewer #1.

Main points:

Point 2: *In this respect, the authors still do not provide any phenotypic information on the DC populations they derive in vitro. As a result, the reader does not know how many of these cells are genuine DC, let alone what proportion are CD103+ or which subset they might belong to.*

We apologize to the reviewer for not addressing this question before (we accidentally only addressed the final part of the original question, which involved primary cells). As requested, we have now extensively characterized the in vitro model for CD103⁺ DCs.

First, we assessed the expression of DC-specific markers. As shown in Suppl. Figure 1, the in vitro cells express DC markers HLA-DR, CD11c, CD83, and CD86. Similar to primary human intestinal DC subsets, CD80 expression is very low¹. In addition, the cells are negative for monocyte/macrophage marker CD14. Finally, nearly all cells stain positive for CD103 (in contrast to monocyte-derived DCs, which are CD103 negative). Combined, these markers indicate that the cells of this in vitro model are indeed CD103 expressing DCs.

Subsequently, we set out to investigate which human CD103⁺ DC subset this in vitro model most closely resembles. For this, we have used the markers of human intestinal DC subsets as identified by Watchmaker et al.¹ The key marker for distinguishing primary human CD103⁺ DC subsets is SIRPα (CD172a). As shown in Suppl. Figure 1, the in vitro CD103⁺ DCs highly express SIRPα, indicating that these cells resemble CD103⁺SIRPα⁺ DCs. To verify this, we assessed additional markers that are differentially expressed by CD103⁺SIRPα⁺ and CD103⁺SIRPα⁻ DCs. Similar to CD103⁺SIRPα⁺ DCs, the in vitro cells express BDCA1 (CD1c) but do not express CLEC9A (in contrast to CD103⁺SIRPα⁻ DCs, which do not express BDCA1 but do express CLEC9A). In addition, similar to CD103⁺SIRPα⁺ DCs, the in vitro cells show intermediate expression of BDCA3 (CD141) and CD83, and show high expression of CD86 (in contrast, CD103⁺SIRPα⁻ show very high BDCA3 expression, are CD83 low, and show relatively low expression of CD86)¹. Please also note that all tested markers appear to be homogeneously expressed (no “double peaks” in the histograms), indicating that the in vitro cells do not contain any major sub-populations.

When combining all these findings, we conclude that the in vitro CD103⁺ DC model most closely resembles human intestinal CD103⁺SIRPα⁺ DCs. As discussed in a recent review by Agace and Mowat, CD103⁺SIRPα⁺ DCs are the subset that mainly mediates Th17 and ILC3 activation², which very nicely corroborates our findings that demonstrate that these cells are able to selectively increase IL-17 production by T helper cells and IL-22 production by intestinal ILC3 (Figure 4).

Point 4: *The expression of FcαR1 on intestinal (from where?) CD103⁺ DC shown in Figure 3a is very low, while the PCR results shown in Figure S4 appear to represent n=1. Given the previous view that DC do not generally express this receptor, more extensive analysis of protein and mRNA by appropriately identified DC subsets and other myeloid cells is needed to allow a full appreciation of the data presented here.*

Indeed, the expression of FcαRI on CD103⁺ DCs is low (which may be an important reason that it has been previously overlooked). As the reviewer indicates, it is insightful to compare FcαRI expression of CD103⁺ DCs with other myeloid cells. Interestingly, when CD103⁺ DCs (Figure 3A is the in vitro model) are compared to in vitro macrophages, which have long been known to express functional FcαRI, FcαRI expression is nearly identical (Suppl. Figure 5). Membrane expression of FcαRI does appear to be higher in primary cells, such as monocytes and Kupffer cells (Suppl. Figure 5).

To study these differences in greater detail we also assessed mRNA expression of various different primary immune cells (Suppl. Figure 6). As expected, neutrophils show the highest relative FcαRI expression, while blood DCs (BDCA1) show the lowest expression. In between are monocytes and the intestinal CD103⁺ DCs.

Taken together, FcαRI expression by CD103⁺ DCs is low, but not much lower than other FcαRI-expressing primary cells (e.g. monocytes) or in vitro models (e.g. macrophages). However, most importantly, FcαRI expression by CD103⁺ DCs is functional, since it dramatically amplifies cytokine production by CD103⁺ DCs, as determined by both using primary cells (Figure 2D) and the in vitro model (e.g. Figure 2A and B).

Other questions:

Point 1: *Although I fully appreciate the difficulties in examining individual subsets of CD103⁺ DC from human intestine, the authors continue to ignore the fact that such subsets exist in the intestine and these are radically distinct in phenotype, function and origin. There is now a substantial literature on these cells and it is not acceptable simply to group "CD103⁺" DC together as an entity for the kind of work presented here.*

We agree with the reviewer that the differences between CD103⁺ subsets was previously underexposed in the manuscript. At the instruction of the editor, we have now clearly acknowledged the diversity of CD103⁺ DC subsets in the discussion of the manuscript, including the references (page 15 line 330). In addition, as mentioned under Point 2, we have now included the extensive characterization of the CD103⁺ DC in vitro model (Suppl. Figure 1).

Point 3: *Despite now providing data on primary intestinal DC in Figure 2D, the amounts of TNFα produced here are very small, no statistics are provided and it is not stated where the "intestinal" samples were obtained from. This last point is important, as there are clear anatomical differences between the subsets of DC found in distinct segments of the large and small intestine.*

- The amounts of TNFα produced by primary cells are indeed small compared to in vitro models. Very likely, this is the result of the harsh and long isolation procedure that is

required to obtain these cells, which includes tissue digestion and cell sorting. Previously, we have also observed this relatively low cytokine production by other primary human cells, e.g. when isolating and sorting cells from human joints³.

- We have now applied statistics to Figure 2D, which with a p value of 0.0517 nearly shows statistical significance.
- As requested, in Figure 2D we have now indicated which segments of the intestine the cells were derived from (1x ileum, 3x colon).

Point 5: *Several figures still lack statistic analysis and the provision of replicate numbers now reveals how few these often are. Of particular note, the data shown in Figure 5C appear to represent n=2, which means that it is invalid to calculate errors and makes the very small differences difficult to interpret.*

Point 6: *In this respect, the data in Figure 5D appear to reflect n=1.*

We generally use the term “n” to indicate the number of independent experiments that were performed using a different donor. When following that definition, n=3 for all panels of Figure 5 (this is a minimum, most experiments were actually done more often). For clarity, we have chosen to show data from one representative example of the different experiments that we have done. The main reason for this is that absolute levels often greatly vary between donors (e.g. see large variation in absolute cytokine levels upon co-stimulation as depicted for multiple donors in Figure 2B). For Figure 5C we tested duplicates (in three different experiments), for time courses such as the data in Figure 5A/B/D we used single values for every individual time point, since this allows us to also study potential differences in kinetics, as previously approved by Nature Communications³.

Point 7: *No studies of signaling molecule expression or function in DC in vivo have been carried out.*

We tried very hard to determine signaling molecule expression in primary human intestinal CD103⁺ DCs, but intracellular staining of these cells was extremely difficult because of high background fluorescence (most likely caused by previous staining for cell sorting). Alternatively, we have determined mRNA expression of relevant FcαRI signaling molecules, which are indeed expressed at similar levels in tested cells (Suppl. Figure 12).

Point 8: *The new data on inhibition of kinase signaling shown in Figure 6H do not show large differences, lack statistical analysis and different scales are used on the separate panels, compromising interpretation.*

Figure 6H was changed at the request of Reviewer #2, together with related textual revision. However, we, as well as the editor, agree with Reviewer #1 that the original figure was more informative (with statistical analysis, same scale, and one only panel). Therefore, we have reverted to the original figure.

Point 9: *How do the authors explain the new findings presented in Figure 6I, which appear to show parallel effects on all parameters assessed, contradicting the earlier findings that TNFα and IL1 may be regulated independently?*

The reviewer raises a very interesting point here: how does fatty acid synthesis amplify IL-1β production, if it does not have any effect on *IL1B* gene translation? We believe that this is mediated

through activation of caspase-1, which has recently been identified to be dependent on fatty acid synthase and is indeed inhibited by inhibitor C75 (the same inhibitor that we used in Figure 6I) ⁴. We thank the reviewer for this point, which we now also have included in the discussion (page 17 line 383) and the graphic summary (Figure 7) .

Combined, we believe that the additional experimental and textual adjustments, particularly regarding the CD103⁺ DC subsets, have even further increased the relevance of the manuscript. We thank Reviewer #1 and editor Dr. Ching-yu Huang for their comments, suggestions, and feedback and look forward to their reply.

Kind regards,

Jeroen den Dunnen (also on behalf of all co-authors)

Reference List

1. Watchmaker PB, *et al.* Comparative transcriptional and functional profiling defines conserved programs of intestinal DC differentiation in humans and mice. *Nat Immunol* **15**, 98-108 (2014).
2. Joeris T, Muller-Luda K, Agace WW, Mowat AM. Diversity and functions of intestinal mononuclear phagocytes. *Mucosal Immunol* **10**, 845-864 (2017).
3. Vogelpoel LT, *et al.* Fc gamma receptor-TLR cross-talk elicits pro-inflammatory cytokine production by human M2 macrophages. *Nat Commun* **5**, 5444 (2014).
4. Moon JS, *et al.* UCP2-induced fatty acid synthase promotes NLRP3 inflammasome activation during sepsis. *J Clin Invest* **125**, 665-680 (2015).

Reviewers' comments:

Reviewer #1 (Remarks to the Author):

I thank the authors for addressing my comments. One or two items remain to be considered:

- 1) While the authors now acknowledge the heterogeneity within CD103+ DC, it would be appropriate that the issue is addressed from the start within the Abstract, Introduction and Results sections, rather than merely as a short comment in the Discussion.
- 2) There is very high expression of CD86 on the in vitro generated DC.
- 3) Supplementary Figure 5 indicates only a very small shift in FcaR expression by the DC and it would be important to see this quantified by eg MFI on replicate samples.
- 4) No positive or negative controls are shown for the PCR data provided in Supplementary Figure 12.

Dear Reviewer #1, dear editor,

We are very excited to hear that, in addition to reviewer #2 and #3, also reviewer #1 is now satisfied with all main points, leaving only a few minor concerns to be addressed. Below, we have made a point-by-point response to these last issues.

- 1) *While the authors now acknowledge the heterogeneity within CD103⁺ DC, it would be appropriate that the issue is addressed from the start within the Abstract, Introduction and Results sections, rather than merely as a short comment in the Discussion.*

As requested, the heterogeneity of the CD103⁺ DCs is now acknowledged in Abstract, Introduction (p. 4, lines 55-59), Results (p. 7, lines 110-117), and Discussion (p. 17, lines 338-349). Please note that we had to be concise, since we reached the maximum word count allowed by Nature Communications (big thanks to the editor for allowing us a few words extra).

- 2) *There is very high expression of CD86 on the in vitro generated DC.*

The high expression of CD86 on (immature) CD103⁺ DCs is indeed interesting. CD86 is commonly used as a maturation marker of DCs, together with CD80 and CD83. Yet, both CD80 and CD83 expression is much lower on the in vitro CD103⁺ DCs. Since primary human CD103⁺SIRP α ⁺ DCs also show elevated expression of CD86 ¹, CD86 may be a potential marker for this subset. Also at the suggestion of the editor, we have implemented this in the text of the manuscript (p. 7, lines 113-117).

- 3) *Supplementary Figure 5 indicates only a very small shift in FcaR expression by the DC and it would be important to see this quantified by eg MFI on replicate samples.*

As requested, we have added a figure depicting the MFI on replicate samples, which shows statistically significant differences (Figure S5B).

- 4) *No positive or negative controls are shown for the PCR data provided in Supplementary Figure 12.*

Please note that this figure is used to show that primary CD103⁺ DCs express the responsible kinases. Both in vitro DC models (CD103⁺ DCs and moDCs) are the positive controls here, since they have long been known to express these kinases. Notably, expression of the kinases is higher in primary CD103⁺ DCs than in the positive controls, demonstrating that primary cells indeed express the responsible signaling molecules. We agree with the reviewer that this was previously unclear, and we have now clearly explained this in the Figure Legend (Figure S12).

Finally, to comply to the editorial policies of Nature Communications, we have shortened the headings in the Results section, which now are all 60 characters or less.

We thank Reviewer #1 and editor Dr. Ching-yu Huang for their comments, suggestions, and feedback and look forward to their reply.

Kind regards,

Jeroen den Dunnen (also on behalf of all co-authors)

1. Watchmaker PB, *et al.* Comparative transcriptional and functional profiling defines conserved programs of intestinal DC differentiation in humans and mice. *Nat Immunol* **15**, 98-108 (2014).

REVIEWERS' COMMENTS:

Reviewer #1 (Remarks to the Author):

I thank the authors for their further changes that have improved the manuscript.